# Increased mitochondrial calcium levels associated with neuronal death in a mouse model of Alzheimer's disease

Maria Calvo-Rodriguez [1], Steven S. Hou[1], Austin C. Snyder[1], Elizabeth K. Kharitonova[1], Alyssa N. Russ[1], Sudeshna Das [1], Zhanyun Fan[1], Alona Muzikansky[2], Monica Garcia-Alloza[3], Alberto Serrano-Pozo[1], Eloise Hudry[1] & Brian J. Bacskai[1 ✉]

Mitochondria contribute to shape intraneuronal $Ca^{2+}$ signals. Excessive $Ca^{2+}$ taken up by mitochondria could lead to cell death. Amyloid beta (Aβ) causes cytosolic $Ca^{2+}$ overload, but the effects of Aβ on mitochondrial $Ca^{2+}$ levels in Alzheimer's disease (AD) remain unclear. Using a ratiometric $Ca^{2+}$ indicator targeted to neuronal mitochondria and intravital multi-photon microscopy, we find increased mitochondrial $Ca^{2+}$ levels associated with plaque deposition and neuronal death in a transgenic mouse model of cerebral β-amyloidosis. Naturally secreted soluble Aβ applied onto the healthy brain increases $Ca^{2+}$ concentration in mitochondria, which is prevented by blockage of the mitochondrial calcium uniporter. RNA-sequencing from post-mortem AD human brains shows downregulation in the expression of mitochondrial influx $Ca^{2+}$ transporter genes, but upregulation in the genes related to mitochondrial $Ca^{2+}$ efflux pathways, suggesting a counteracting effect to avoid $Ca^{2+}$ overload. We propose lowering neuronal mitochondrial $Ca^{2+}$ by inhibiting the mitochondrial $Ca^{2+}$ uniporter as a novel potential therapeutic target against AD.

[1] Department of Neurology, Massachusetts General Hospital and Harvard Medical School, 114, 16th St, Charlestown, MA 02129, USA. [2] Department of Biostatistics, Harvard School of Public Health, 50 Staniford Street, Boston, MA, USA. [3] Division of Physiology, School of Medicine, Instituto de Investigacion Biomedica de Cadiz (INIBICA), Universidad de Cadiz, Cadiz, Spain. ✉email: BBACSKAI@mgh.harvard.edu

Alzheimer's disease (AD), the most common age-related neurodegenerative disorder, is characterized by deposition of extracellular senile plaques, neurofibrillary tangles and neurodegeneration, which lead to cognitive impairment and dementia. Despite the extensive efforts to understand its pathophysiology, the exact mechanisms underlying AD remain unknown. The most influential hypothesis for AD postulates that abnormal accumulation of the amyloid β (Aβ) peptide is the precipitating event that triggers a pathogenic cascade of adverse events, and that Aβ oligomers (Aβo), rather than amyloid plaques themselves, are the neurotoxic species of Aβ[1] linked to tau aggregation[2]. Aβ neurotoxicity has also been associated with intraneuronal $Ca^{2+}$ dyshomeostasis[3]. The calcium hypothesis of AD[4] posits that activation of the amyloidogenic pathway remodels neuronal $Ca^{2+}$ signaling, altering normal $Ca^{2+}$ homeostasis and the mechanisms responsible for learning and memory. Previous work from our group and others has shown that Aβ aggregates increase cytosolic $Ca^{2+}$ levels in vivo both in a mouse model of AD[5,6], and after application of soluble Aβo to the brain of naive wild-type (Wt) mice[7,8]. However, these investigations assessed $Ca^{2+}$ levels in the cytosol, and whether Aβ affects $Ca^{2+}$ levels in mitochondria in vivo remains elusive.

Mitochondrial malfunction has been proposed as an early event in AD and other aging-related neurodegenerative disorders[9]. Studies on brains from AD patients and AD mouse models have shown impaired mitochondrial function, manifested as decreased bioenergetics and ATP synthesis[10], morphological abnormalities[11], imbalance of mitochondrial dynamics[12] and redistribution of mitochondria[13]. Mitochondria participate in intracellular $Ca^{2+}$ signaling as modulators, buffers and sensors[14]. After a cytosolic $Ca^{2+}$ increase, mitochondria rapidly take up $Ca^{2+}$ to prevent $Ca^{2+}$ overload into the cytosol. However, excessive $Ca^{2+}$ taken up into mitochondria, i.e., mitochondrial $Ca^{2+}$ overload, results in increased reactive oxidative species (ROS) production, ATP synthesis inhibition, mitochondrial permeability transition pore (mPTP) opening, release of cytochrome c, activation of caspases and apoptosis[15]. In vitro studies have reported that Aβo induce mitochondrial $Ca^{2+}$ uptake[16,17], and $Ca^{2+}$ transfer from ER to mitochondria[18] in cultured rat primary neurons, but whether these findings apply in vivo remains unknown. In the present study, we assess whether Aβ aggregates alter mitochondrial $Ca^{2+}$ levels in neurons in the mouse brain in vivo. We find elevated $Ca^{2+}$ levels in neuronal mitochondria in transgenic mice, but only after plaque deposition, and preceding neural death. This mitochondrial $Ca^{2+}$ overload in vivo involves toxic extracellular Aβ oligomers and requires the mitochondrial $Ca^{2+}$ uniporter. We propose reducing mitochondrial $Ca^{2+}$ overload by blocking the mitochondrial $Ca^{2+}$ uniporter as a novel potential therapeutic target for AD.

## Results

### Functional AAV.hSyn.2mtYC3.6 targets neuronal mitochondria.
We have previously observed cytosolic $Ca^{2+}$ overload[5] and mitochondrial membrane potential decrease[11] in the APPswe/PSEN1ΔE9 (APP/PS1) Tg mouse after amyloid plaque deposition. Here we hypothesized that Aβ aggregates could also increase neuronal mitochondrial $Ca^{2+}$ levels in vivo. To study free $Ca^{2+}$ concentration in mitochondria ($[Ca^{2+}]_{mit}$) in neurons, we targeted the expression of the ratiometric $Ca^{2+}$ indicator Yellow Cameleon 3.6 (YC3.6)[19] to neuronal mitochondria (Fig. 1a). YC3.6 allows visualization of subcellular $Ca^{2+}$ concentration based on Förster resonance energy transfer (FRET) imaging between CFP and YFP, quantitatively measuring absolute $Ca^{2+}$ concentrations. YC3.6 is one of the brightest reporters among the YC versions, and exhibits a large dynamic range and excellent

signal-to-noise ratio[19]. Detailed characterization of hSyn.2mtYC3.6 expression was performed in vitro using neuroblastoma Neuro2a cells (N2a) and mouse primary cortical neurons, and ex vivo after immunofluorescence of sections from a Wt adult mouse brain injected with AAV.hSyn.2mtYC3.6. The punctate and perinuclear pattern expression of 2mtYC3.6 (Fig. 1b, c) suggested its mitochondrial localization. Co-transfection of cells with hSyn.2mtYC3.6 and mRuby-Mito-7 confirmed mitochondrial colocalization of 2mtYC3.6 (Fig. 1b). Co-labeling of AAV.hSyn.2mtYC3.6 with HSP60 antibody (Heat shock protein 60, localized to the mitochondrial matrix) in tissue sections corroborated the mitochondrial and neuronal-specific expression of the 2mtYC3.6 in vivo (Fig. 1c and Supplementary Fig. 1).

We calibrated hsyn.2mtYC3.6 in N2a cells. Cells were transfected with hSyn.2mtYC3.6, and 24 h later, cells were permeabilized and exposed to solutions containing increasing known $[Ca^{2+}]$ (Methods). An increase in the YFP/CFP fluorescence ratio of YC3.6 corresponds to an increase in $[Ca^{2+}]$. Ratios of YFP/CFP fluorescence intensities were plotted against $[Ca^{2+}]$ and fitted with a sigmoidal curve (Fig. 1d). We observed nearly 4-fold increase in the ratio at high $[Ca^{2+}]_{mit}$ ($R_{min}=0.606$, $R_{max} = 2.6921$), with an apparent $K_d$ ($K_d'$) of 4.21 μM and a Hill coefficient of 1.57[20]. Either the mitochondrial environment or the fusion of two mitochondrial sequences shifted the affinity of YC3.6 for $Ca^{2+}$ to higher values, making it more suitable for $Ca^{2+}$ measurements in mitochondria. This calibration was used to obtain $[Ca^{2+}]_{mit}$ in the subsequent experiments. To verify that AAV.hSyn.2mtYC3.6 was functional in vivo, we applied KCl topically onto the brain of C57BL/6J mice injected with AAV.hSyn.2mtYC3.6 to induce neuronal depolarization, and imaged the neuronal mitochondria with multiphoton microscopy. Figure 1e and Supplementary Video 1 show an increase in the YFP/CFP ratio, representing $[Ca^{2+}]_{mit}$ (pseudocolored images and $Ca^{2+}$ recording), immediately after KCl application. Therefore, AAV.hSyn.2mtYC3.6 specifically targets mitochondria in neurons, can be expressed in vivo and provides a quantitative readout of neuronal $[Ca^{2+}]_{mit}$.

We compared the kinetics of cytosolic $Ca^{2+}$ versus mitochondrial $Ca^{2+}$ in primary neurons upon KCl-induced depolarization. We loaded cortical neurons with the cytosolic $Ca^{2+}$ indicator calbryte 590, and infected them with our mitochondrial reporter hSyn.2mtYC3.6, to measure $Ca^{2+}$ changes in the cytosol and mitochondria simultaneously (Supplementary Fig. 2). Application of KCl increased both cytosolic $Ca^{2+}$ concentration ($[Ca^{2+}]_{cyt}$) and $[Ca^{2+}]_{mit}$. However, mitochondria exhibit different kinetics, which justified the need to address the mitochondrial $Ca^{2+}$ impairment independent of cytosolic $Ca^{2+}$ in the disease model.

### Mitochondrial $Ca^{2+}$ overload in vivo after Aβ plaque deposition.
To address whether there are alterations in mitochondrial $Ca^{2+}$ contributing to a general $Ca^{2+}$ dyshomeostasis in AD, we examined $[Ca^{2+}]_{mit}$ in the APP/PS1 Tg mouse model of cerebral β-amyloidosis in vivo using multiphoton microscopy. Age-matched Wt littermates were used as controls. 8-months-old (mo-old) APP/PS1 Tg mice were intracortically injected with AAV.hSyn.2mtYC3.6 targeting layers II/III, and 5 mm diameter cranial windows were implanted over the somatosensory cortex. Imaging was conducted 3 weeks later to allow for protein expression. Methoxy X0-4 was injected intraperitoneally 24 h before to label amyloid plaques[21] (Fig. 2a). $[Ca^{2+}]_{mit}$ was assessed by acquiring volumes of different regions of the cortex and determining the YFP/CFP ratio in the individual mitochondria. Figure 2b shows in vivo multiphoton microscopy images of mitochondria in neurons after transduction by AAV.hSyn.2mtYC3.6 in 9-mo-old Wt (top panel) and APP/PS1 Tg mice

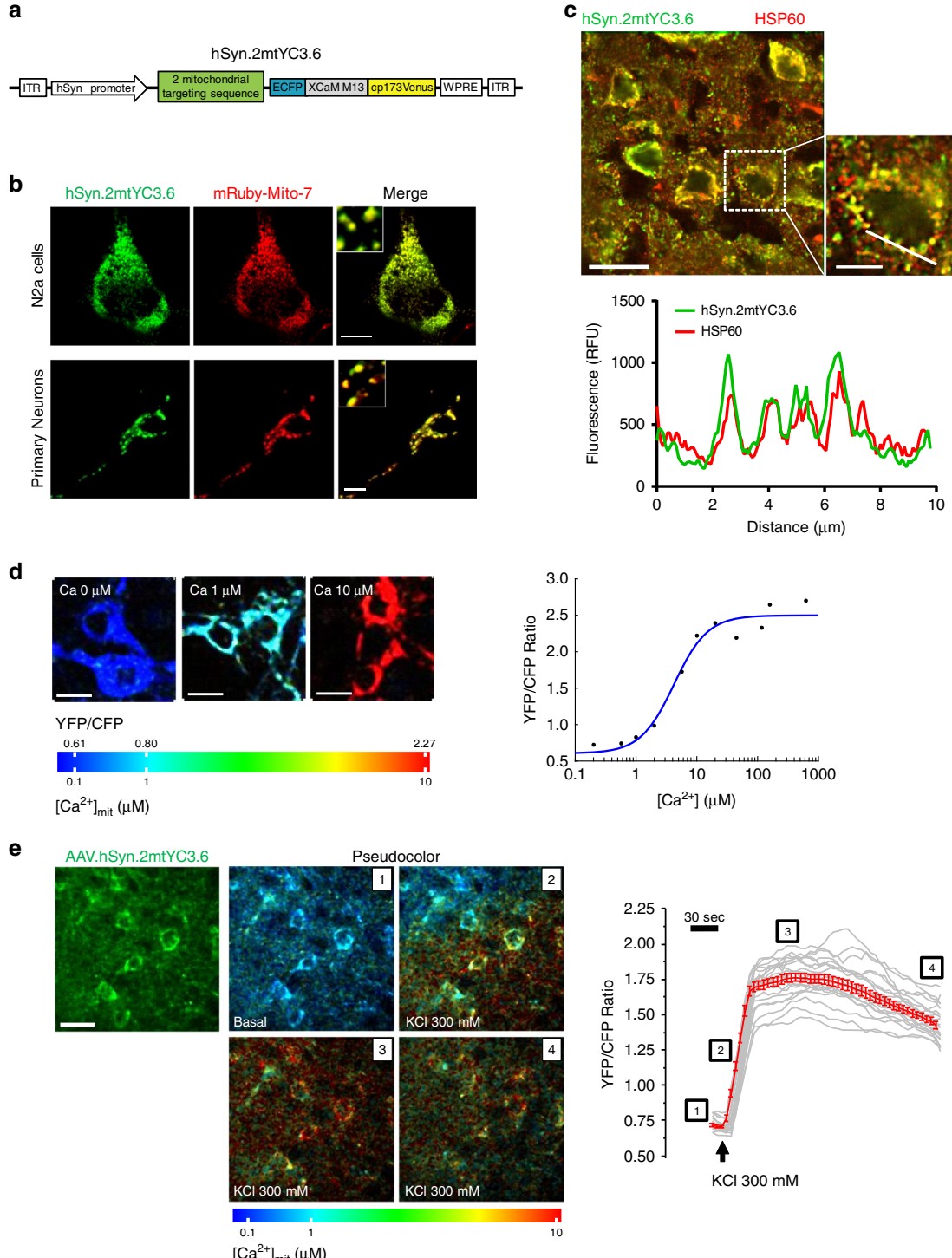

**Fig. 1 hSyn.2mtYC3.6 targets neuronal mitochondria and is functional in vivo. a** Diagram of construct AAV.hSyn.2mtYC3.6-WPRE. **b** Mitochondrial counter labeling of 2mtYC3.6 verified proper targeting to mitochondria. N2a cells (top) and primary cortical neurons (bottom) were co-transfected with hSyn.2mtYC3.6 (green) and mRuby-Mito-7 (red) and subjected to confocal microscopy imaging. Scale bar 10 μm. **c** Colocalization of HSP60 (red) and hSyn.2mtYC3.6 (green) in cortex shown by immunohistochemistry (top). Scale bar 20 μm. Bottom graph shows intensity profile of the ROI across the cell (inset, scale bar 10 μm). Green line represents the intensity of hSyn.2mtYC3.6 and red line represent the intensity of HSP60. **d** In vitro calibration of 2mtYC3.6. Primary cortical neurons and N2a cells were exposed to varying [Ca$^{2+}$] solutions, and relative changes in YFP/CFP ratio vs. [Ca$^{2+}$] are indicated. Left, images are pseudocolored according to the color bar. Scale bar 10 μm. Right, titration curve of 2mtYC3.6 in N2a cells, two independent experiments, number of cells: 0 μM, $n = 233$; 0.2 μM, $n = 339$; 0.575 μM, $n = 436$; 1 μM, $n = 332$; 2 μM, $n = 321$; 5.7 μM, $n = 356$; 10 μM, $n = 295$; 20 μM, $n = 317$; 53 μM, $n = 372$; 100 μM, $n = 372$; 200 μM, $n = 292$; 500 μM, $n = 209$; 1000 μM, $n = 274$. **e** Validation of AAV.hSyn.2mtYC3.6 in vivo. Left, multiphoton microscopy imaging of the expression of AAV.hSyn.2mtYC3.6 in the cortex of a C57BL/6 J mouse. Scale bar 20 μm. Middle, pseudocolored images of the mitochondria expressing AAV.hSyn.2mtYC3.6 in basal conditions and after KCl application. Right, recording of the application of KCl to the brain. Gray traces represent single-cell responses and red trace represents averaged (mean ± SEM) [Ca$^{2+}$]$_{mit}$ responses to 300 mM KCl. Representative of three mice.

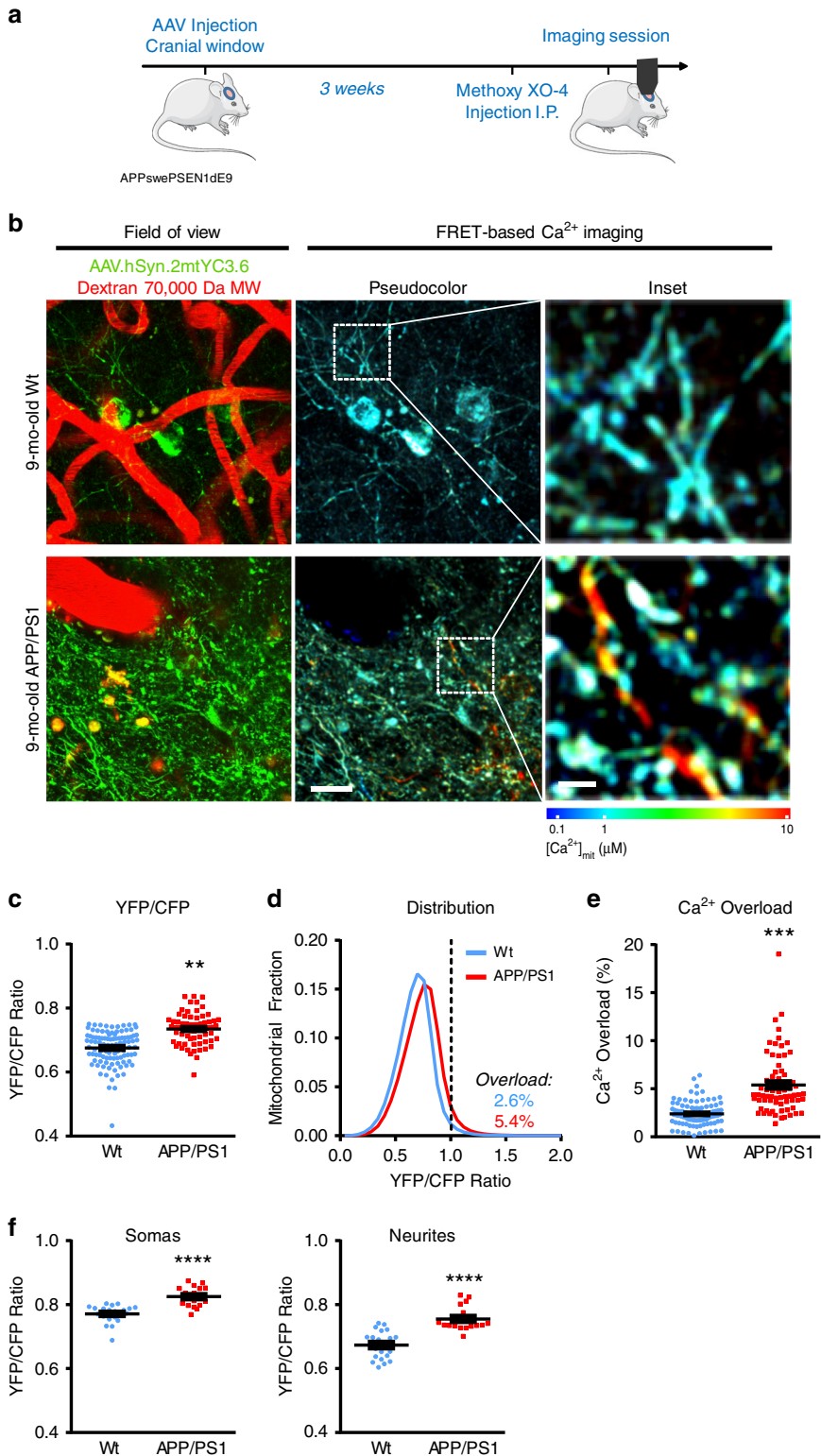

(bottom panel). On average, 9-mo-old APP/PS1 Tg mice exhibited higher YFP/CFP ratios than 9-mo-old Wt mice (Fig. 2c).We also evaluated mitochondrial $Ca^{2+}$ overload using frequency distribution histograms (Fig. 2d). A general shift towards higher YFP/CFP ratios (higher $[Ca^{2+}]_{mit}$) was observed for APP/PS1 Tg mice. $Ca^{2+}$ overload was defined as YFP/CFP ratio greater than two standard deviations (SD) above the mean YFP/CFP ratio obtained for all the mitochondria in all Wt mice[5,7,22] (dashed line, YFP/CFP ratio >0.99, ~1637 nM $[Ca^{2+}]_{mit}$). Twice as many mitochondria from APP/PS1 Tg mice showed $Ca^{2+}$ overload compared to Wt mice (Fig. 2e). $[Ca^{2+}]_{mit}$ was also higher in the APP/PS1 Tg mice in both somas and neurites (Fig. 2f).

Next, we assessed whether the presence of Aβ plaques was required for the increased resting $[Ca^{2+}]_{mit}$, or whether it

**Fig. 2 Mitochondrial Ca$^{2+}$ overload after Aβ plaque deposition. a** Experimental procedure to determine [Ca$^{2+}$]$_{mit}$ in neurons in APP/PS1 Tg mice. 8-mo-old mice were injected with AAV.hSyn.2mtYC3.6 and a cranial window was implanted. Three weeks later, [Ca$^{2+}$]$_{mit}$ was assessed by multiphoton microscopy. **b** In vivo multiphoton microscopy images of mitochondria expressing AAV.hSyn.2mtYC3.6 in Wt (top) and APP/PS1 Tg mice (bottom). Field of view shows AAV.hSyn.2mtYC3.6 in mitochondria (green), and blood vessels labeled with Dextran 70,000 MW (red). FRET-based Ca$^{2+}$ imaging shows pseudocolored images of the field of view (scale bar 20 μm) and the inset from the white box (scale bar 5 μm). **c** Nine-mo-old APP/PS1 Tg mice exhibited increased ration YFP/CFP when compared to 9-mo-old Wt mice. Scatter dot plot representing the ratio YFP/CFP of every volume acquired. Error bars represent mean ± SEM. **\*\*p = 0.0015 (0.67 ± 0.011 (~472 nM [Ca$^{2+}$]$_{mit}$), n = 100 z-stacks from 11 Wt mice, and 0.73 ± 0.014 (~731 nM [Ca$^{2+}$]$_{mit}$), n = 70 z-stacks from 7 APP/PS1 Tg mice). **d** Histogram of [Ca$^{2+}$]$_{mit}$ frequency distribution (indicated by YFP/CFP ratio). The black dotted line corresponds to 2 SD above the mean [Ca$^{2+}$]$_{mit}$ in Wt mice (YFP/CFP ratio >0.99, ~1637 nM). Ratios above this line were classified as mitochondrial Ca$^{2+}$ overload. **e** Comparison of the percentage of Ca$^{2+}$-overloaded mitochondria in 9-mo-old Wt and APP/PS1 Tg mice. APP/PS1 Tg mice exhibited significantly higher percentage of mitochondria with elevated Ca$^{2+}$. Scatter dot plot represents the percentage of Ca$^{2+}$-overloaded mitochondria of every volume acquired. Error bars represent mean ± SEM. **\*\*\*p = 0.001 (5.4 ± 0.79% from n = 70 z-stacks from 7 APP/PS1 Tg mice vs. 2.6 ± 0.51% from n = 100 z-stacks from 11 Wt mice). **f** Comparison of [Ca$^{2+}$]$_{mit}$ (YFP/CFP ratio) in the different compartments (somas and neurites) in 9-mo-old Wt and APP/PS1 Tg mice. APP/PS1 Tg mice exhibited significantly higher Ca$^{2+}$ levels in mitochondria both in somas and neurites. Error bars represent mean ± SEM. **\*\*\*\*p < 0.0001 (Somas: 0.77 ± 0.01, n = 21 z-stacks from 3 Wt mice, and 0.82 ± 0.01, n = 18 z-stacks from 3 APP/PS1 Tg mice. Neurites: 0.67 ± 0.01, n = 21 z-stacks from 3 Wt mice, and 0.76 ± 0.01, n = 18 z-stacks from 3 APP/PS1 Tg mice). **a** Created using modified Servier Medical Art templates, which are licensed under a Creative Commons Attribution 3.0 Unported License (https://creativecommons.org/licenses/by/3.0/).

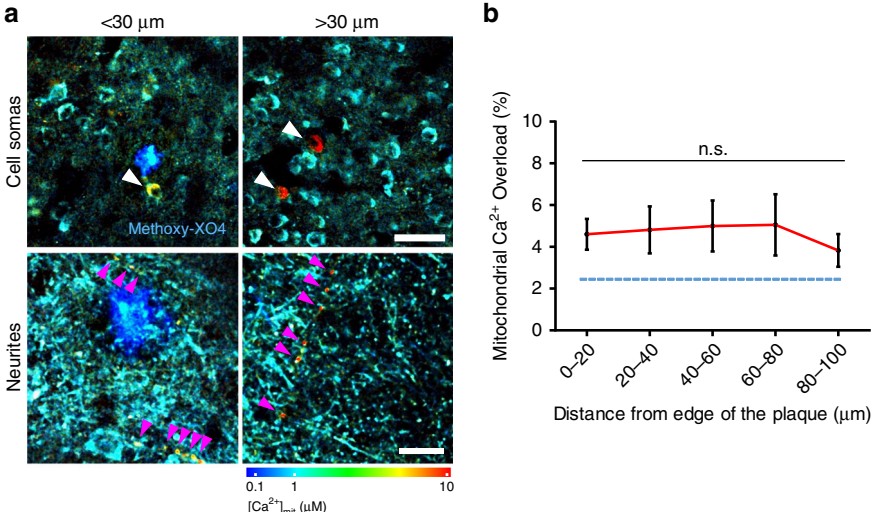

**Fig. 3 Mitochondrial Ca$^{2+}$ overload is independent of the distance to plaques. a** In vivo multiphoton microscopy pseudocolored images of AAV. hSyn.2mtYC3.6 expressing mitochondria in 9-mo-old APP/PS1 Tg mice. Elevated [Ca$^{2+}$]$_{mit}$ was observed in mitochondria at <30 μm distance (left), but also at >30 μm distance (right). Arrow heads show mitochondria with Ca$^{2+}$ overload in neuronal soma (top, scale bar 50 μm) and neurites (bottom, scale bar 10 μm). **b** Within 100 μm distance from the edge of a plaque, the probability of finding mitochondrial Ca$^{2+}$ overload did not change. n = 31 plaques analyzed from 5 Tg mice (mean ± SEM, n.s. non-significant). The percentage of mitochondria showing Ca$^{2+}$ overload in Wt mice is shown as reference (blue dotted line).

preceded plaque formation. APP/PS1 Tg mice start developing amyloid plaques at 4–5 mo-old[23]. No differences in YFP/CFP ratio or Ca$^{2+}$ overload were detected between Wt and APP/PS1 mice at 2.5–3-mo-old (before plaque deposition), in either somas or neurites (Supplementary Fig. 3). Histograms in Supplementary Fig. 3 show that at this age there is no shift toward higher YFP/CFP ratios in APP/PS1 Tg mice relative to Wt mice at this early age. This is consistent with previous studies reporting no changes in [Ca$^{2+}$]$_{cyt}$ and neuronal function prior to plaque deposition[5,24,25], supporting the idea that mitochondrial Ca$^{2+}$ overload in neurons follows, rather than precedes, amyloid plaque deposition.

**Mitochondrial Ca$^{2+}$ is independent from the distance to plaques.** Individual Aβ plaques were previously reported to increase the likelihood of Ca$^{2+}$ overload in the cytosol of neurites and cell bodies within a range of ~20–30 μm from the plaque edge[5,6] in

APP/PS1 Tg mice. To test whether Aβ plaques have local effects on the [Ca$^{2+}$]$_{mit}$ in these mice, we measured mitochondrial Ca$^{2+}$ overload relative to the distance between each mitochondrion and the edge of the nearest amyloid plaque. No significant differences were found in the average mitochondrial Ca$^{2+}$ overload as a function of distance to plaques in either cell bodies or neurites (Fig. 3a, b), suggesting that soluble Aβo species, rather than Aβ plaques per se, are likely responsible for the increase of neuronal [Ca$^{2+}$]$_{mit}$. Indeed, it has been shown that the concentration of soluble Aβo in the brain of this mouse model increases with age[23,26,27], which could explain the observed age-related [Ca$^{2+}$]$_{mit}$ increase.

**Soluble Aβ increases mitochondrial Ca$^{2+}$ in Wt mice neurons.** We have previously shown that acute exposure of the brain of naive (Wt) living mice to soluble Aβo leads to disruption of cytosolic Ca$^{2+}$ homeostasis in neurons[7]. To test the hypothesis

that soluble Aβ is responsible for the $[Ca^{2+}]_{mit}$ increase, we topically applied naturally secreted soluble Aβo onto the brain of Wt mice expressing AAV.hSyn.2mtYC3.6, and assessed $[Ca^{2+}]_{mit}$ with multiphoton microscopy. Aβo-enriched medium was obtained from primary cortical neurons isolated from Tg2576 embryos (APP Swedish mutation) (Tg-conditioned media, TgCM), which mainly contains low molecular weight oligomers (dimers to tetramers)[7,28]. CM from Wt-littermate neurons (WtCM) and Aβ-immunodepleted TgCM were used as controls. The three media were subjected to an $Aβ_{40}$ sandwich ELISA. $Aβ_{40}$ concentration in the media was 0.3, 5 and 0.7 nM for WtCM, TgCM and Aβ-immunodepleted TgCM, respectively. Aβo concentration in TgCM represents around 10% of the total amount of $Aβ_{40}$[7].

To assess the effects of CM on $[Ca^{2+}]_{mit}$ in the healthy brain, a cranial window was implanted over the somatosensory cortex of 4- to 5-mo-old C57BL/6J mice injected with AAV.hSyn.2mtYC3.6, the mouse was imaged at baseline, the window was reopened to apply either WtCM, TgCM or Aβ-depleted TgCM onto the brain for 1 h, and finally, the mouse was reimaged in the same volumes imaged at baseline (Fig. 4a, b). Mitochondrial $Ca^{2+}$ overload was defined as the YFP/CFP ratio greater than 2SD above the mean YFP/CFP ratio of all the mitochondria imaged under basal conditions (YFP/CFP ratio >0.98, (~1604 nM $[Ca^{2+}]_{mit}$)). Application of TgCM significantly increased $[Ca^{2+}]_{mit}$, causing a shift towards higher YFP/CFP ratios (higher $[Ca^{2+}]_{mit}$) in the $[Ca^{2+}]_{mit}$ histogram (Fig. 4c, middle). The proportion of mitochondria with $Ca^{2+}$ overload after application of TgCM was significantly different than in basal conditions (Fig. 4d, middle). By contrast, topical application of WtCM did not increase $[Ca^{2+}]_{mit}$ (Fig. 4c, left). To further validate that Aβo are responsible for the mitochondrial $Ca^{2+}$ overload, we immunodepleted human Aβ from the TgCM with an anti-β-Amyloid antibody (β-amyloid 1–16, 6E10). Per ELISA measurements, immunodepletion of Aβ effectively removed about 85% of Aβ present in TgCM, from 5 to 0.7 nM (final $Aβ_{40}$ concentration). Application of Aβ-immunodepleted TgCM did not increase $[Ca^{2+}]_{mit}$ (Fig. 4c, right) compared to basal conditions. Comparing the averaged YFP/CFP ratios in each volume (from all mitochondria in each z-stack) acquired before and after application of CM (Fig. 4e), the majority of z-stacks analyzed after the application of TgCM were increased by 5% or more (dark lines), whereas only a few z-stacks showed a ≥5% increase for WtCM and Aβ-immunodepleted TgCM. Using these ratios, we calculated the relative change in YFP/CFP ratio ($ΔR/R_0$) for each z-stack. Application of TgCM induced a relative change in the YFP/CFP ratio significantly higher than WtCM or Aβ-immunodepleted TgCM (Fig. 4f). This effect was more prominent in neurites (Fig. 4g). These relatively small, but significant differences, suggest that a small fraction of neuronal mitochondria are affected by TgCM, similar to our observation in plaque-bearing APP/PS1 Tg mice, and implicate of soluble Aβo in mitochondrial $Ca^{2+}$ overload in neurons in vivo.

**MCU is required for Aβo-driven mitochondrial $Ca^{2+}$ uptake.** The main $Ca^{2+}$ uptake pathway in mitochondria is the mitochondrial $Ca^{2+}$ uniporter (MCU) complex, which includes the MCU pore, located in the inner mitochondrial membrane[29,30], and the combination of two regulatory subunits MiCU1 and MiCU2 (mitochondrial calcium uptake 1 and 2 proteins), which regulate the MCU response to extra-mitochondrial $Ca^{2+}$[31,32]. Other proteins involved in the complex include the essential MCU regulator (EMRE)[33], MCUb[34], or MiCU3[35]. Mitochondrial $Ca^{2+}$ efflux depends mainly on the electrogenic $Na^+$-dependent $Ca^{2+}$ release exchanger (NCLX)[36]. Given the increased number

of mitochondria with $Ca^{2+}$ overload in APP/PS1 Tg mice after amyloid plaque deposition, we performed Western blots for MCU, MiCU1, MiCU2, and NCLX in mitochondrial-enriched fractions of the brain of 10-mo-old Wt and APP/PS1 Tg mice to test whether Aβ affects the expression levels of these proteins. No significant differences were detected between genotypes in any of these mitochondrial proteins (Supplementary Fig. 4).

To test whether Aβo induce mitochondrial $Ca^{2+}$ increase in neurons via the MCU complex, we blocked the pore with the specific cell-permeable MCU inhibitor Ruthenium 360 (Ru360)[37]. As in Fig. 4a, C57BL/6J Wt mice expressing hSyn.2mtYC3.6 were subjected to a craniotomy, and $[Ca^{2+}]_{mit}$ was assessed in basal conditions. The window was then reopened and the underlying brain was pre-treated with Ru360 (100 μM) for 15 min. TgCM mixed with Ru360 was next applied onto the brain for 1 h, and the mouse was reimaged in the same areas. TgCM increased neuronal $[Ca^{2+}]_{mit}$ (arrow heads) as expected, but application of TgCM mixed with Ru360 did not (Fig. 5a). Pre-treatment with Ru360 did not shift the distribution of YFP/CFP ratios when compared to TgCM alone (Fig. 5b), and the proportion of mitochondria with $Ca^{2+}$ overload, (YFP/CFP ratio >1.01 (~1704 nM $[Ca^{2+}]_{mit}$)), was not significantly different (Fig. 5c). Topical application of Ru360 alone did not have an effect on $[Ca^{2+}]_{mit}$. Comparing the average YFP/CFP ratio in each volume acquired before and after application of TgCM with Ru360, only a few volumes increased 5% or more (dark lines), whereas for the majority the increase was ≤5% (light lines) (Fig. 5d). Average of the relative change in YFP/CFP ratio ($ΔR/R_0$) for each volume (Fig. 5e) showed that application of TgCM without Ru360 induced a change in YFP/CFP ratio significantly different from the application of TgCM with Ru360. Importantly, Ru360 application did not inhibit the rise of $Ca^{2+}$ in the cytosol induced by TgCM (as tested in C57Bl/6J mice injected with CBA. YC3.6[5]) (Supplementary Fig. 5), suggesting that Ru360 did not render neurons insensitive to Aβo or blocked channels/influx pathways in the plasma membrane. These data suggest that MCU is required for the Aβ-driven mitochondrial $Ca^{2+}$ increase in neurons in the brain in vivo.

**Mitochondrial $Ca^{2+}$ precedes neuronal death.** We next asked whether the excessive $Ca^{2+}$ could have a detrimental effect in $Ca^{2+}$ overloaded mitochondria and, eventually, also in the neurons containing these mitochondria. We examined the time course of Aβo-driven neuronal mitochondrial $Ca^{2+}$ overload in the C57BL/6 J Wt mouse brain, and specifically asked whether this was permanent or reversible and, if the latter, how long after Aβo application. At 8 h post-application of TgCM, neuronal $[Ca^{2+}]_{mit}$ had already returned to basal levels (Fig. 6d–e). As expected, WtCM had no effect on $[Ca^{2+}]_{mit}$ (Fig. 6a–c). Despite the relatively transient mitochondrial $Ca^{2+}$ overload associated to acute topical application of Aβo, we noted altered mitochondria shape and size at 8 h only with TgCM (but not with WtCM) compared to basal conditions (Fig. 6a, d, insets; Supplementary Figs. 6 and 7), in agreement with previous reports showing mitochondrial morphological abnormalities in the human AD brain and AD Tg mice[11,38].

Because mitochondrial $Ca^{2+}$ overload can lead to apoptosis[39], we examined the viability of neurons with chronic (rather than transient) mitochondrial $Ca^{2+}$ overload using longitudinal multiphoton microscopy serial imaging over a 24–48 h period in 9-mo-old APP/PS1 Tg mice. First, we evaluated whether the static mitochondrial $Ca^{2+}$ levels were stable longitudinally. We observed that over 3 imaging sessions (48 h), the absolute mitochondrial $Ca^{2+}$ levels were constant in the same cell, for both low and intermediate $Ca^{2+}$ levels (Fig. 7a). Surprisingly, a

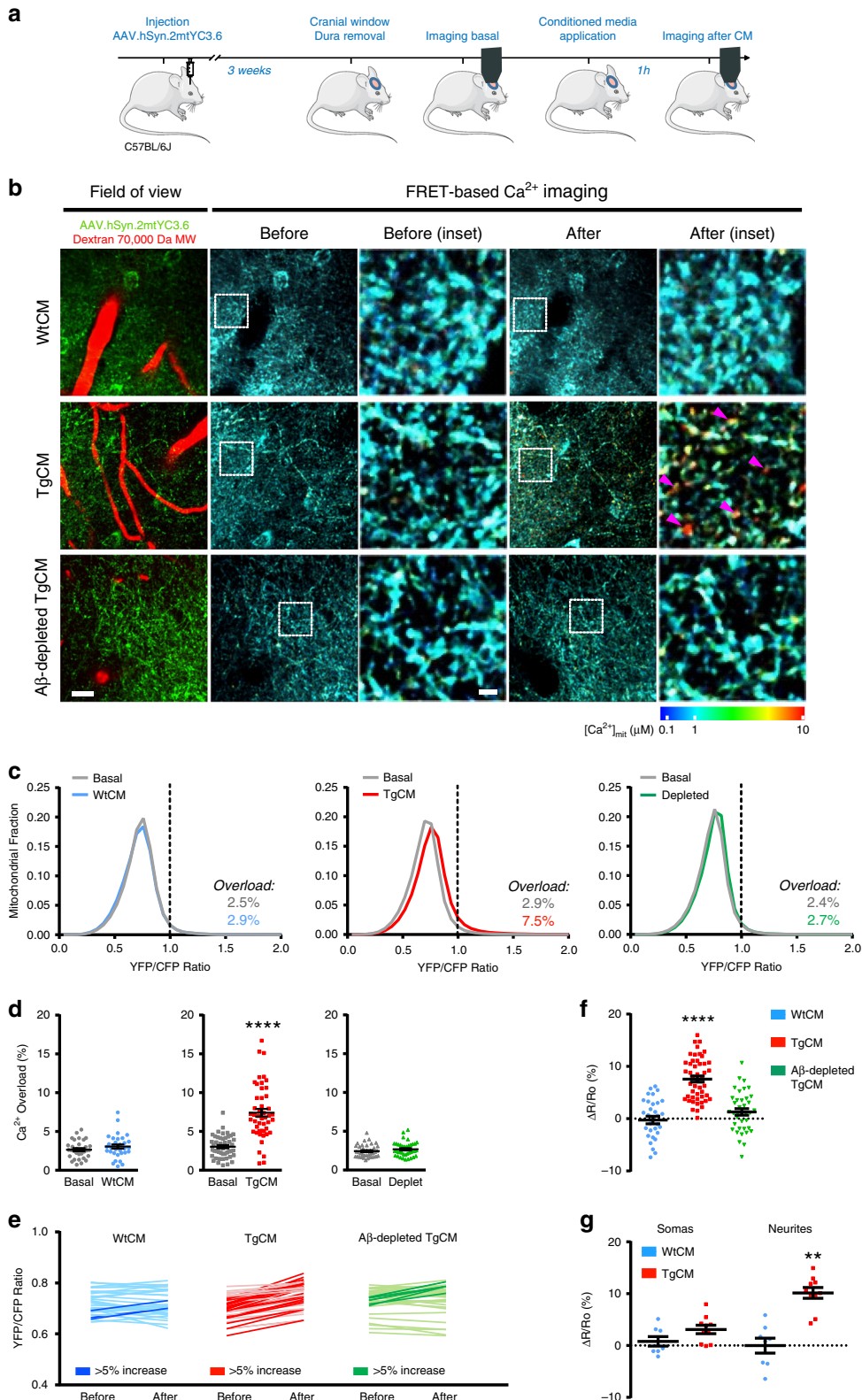

small fraction of neurons with extremely high $[Ca^{2+}]_{mit}$ in their somas at the first imaging session (on average, YFP/CFP ratio ≥1.4, (~3093 nM $[Ca^{2+}]_{mit}$)) were lost within 24 h (Fig. 7b, c). None of the neurons exhibiting YFP/CFP ratio ≤1.4 disappeared in the second imaging session. We concomitantly injected Hoechst 33342 with hSyn.2mtYC3.6 to observe the nucleus and $[Ca^{2+}]_{mit}$ simultaneously. We observed that the nucleus from

healthy neurons was spherical and chromatin was evenly distributed. Conversely, in neurons with high $[Ca^{2+}]_{mit}$ at baseline, the nucleus became condensed (likely undergoing apoptosis), and the soma lost its of YFP/CFP fluorescence (Fig. 7c). Despite being an occasional event in this model[40], this observation suggests a close correlation between high $[Ca^{2+}]_{mit}$ and neuronal cell death.

**Fig. 4 Soluble Aβ increases mitochondrial Ca$^{2+}$ concentration in neurons in vivo. a** Experimental procedure: 4-mo-old C57BL/6J mice were injected with AAV.hSyn.2mtYC3.6. A cranial window was implanted the day of the experiment (3 weeks later). [Ca$^{2+}$]$_{mit}$ was assessed in basal conditions. Then, the window was reopened and either WtCM, TgCM or Aβ-immunodepleted TgCM was applied for 1 h. Finally, the mouse was reimaged in the exact same regions as for basal conditions. **b** Representative pictures of the effects of CM in the healthy living mouse brain. Arrow heads show mitochondria with high [Ca$^{2+}$]. Scale bar 20 and 5 μm for the inset. **c** Histograms of [Ca$^{2+}$]$_{mit}$ frequency distribution (YFP/CFP ratio) for the three conditions before (basal) and after application of CM. The black dotted line corresponds to 2SD above the mean [Ca$^{2+}$]$_{mit}$ measured for all mice in baseline. Ratios above this line were classified as mitochondrial Ca$^{2+}$ overload. **d** Scatter dot plot represents the percentage of mitochondria showing Ca$^{2+}$ overload before and after application of CM for every volume acquired. Error bars represent mean ± SEM (WtCM: before 2.45 ± 0.43% vs. after 2.92 ± 0.49%; non-significant, $n =$ 32 z-stacks from 5 mice. TgCM: before 2.89 ± 0.47% vs. after 7.53 ± 1.09%; ****$p < 0.0001$, $n =$ 48 z-stacks from 7 mice. Aβ-depleted TgCM: before 2.43 ± 0.13% vs. after 2.66 ± 0.20%; non-significant, $n =$ 38 from z-stacks 5 mice). The percentage of Ca$^{2+}$ overload before (basal) and after CM application is also noted on the graphs. **e** Averaged YFP/CFP ratios before and after treatment for each z-stack acquired. The dark traces account for the z-stacks showing an increase ≥5% in YFP/CFP ratio after topical application of CM. **f** Scatter dot plot represents the relative change in ratio for each condition. Error bars represent mean ± SEM (WtCM: 0.20 ± 1.09%, $n =$ 32 z-stacks; TgCM: 7.59 ± 0.92%, 48 z-stacks; depleted TgCM 1.20 ± 1.05%, 38 z-stacks from 5, 7 and 5 mice, respectively, ****$p < 0.0001$). **g** Relative increase of [Ca$^{2+}$]$_{mit}$ (ΔR/R$_0$) in the different compartments (somas and neurites) after WtCM or TgCM application. Error bars represent mean ± SEM. **$p < 0.005$ (Somas: 0.79 ± 0.91, $n =$ 8 z-stacks from 3 WtCM application mice, and 3.10 ± 0.82, $n =$ 10 z-stacks from 3 TgCM application mice. Neurites: −0.02 ± 1.44, $n =$ 8 z-stacks from 3 WtCM application mice, and 10.15 ± 1.05, $n =$ 10 z-stacks from 3 TgCM application mice). **a** Created using modified Servier Medical Art templates, which are licensed under a Creative Commons Attribution 3.0 Unported License (https://creativecommons.org/licenses/by/3.0/).

To further understand the effects of increased mitochondrial Ca$^{2+}$ concentration and the connection to neuronal death, we performed some mechanistic studies in vitro using mouse primary cortical neurons exposed to TgCM. First, we determined that TgCM increased [Ca$^{2+}$]$_{mit}$ in primary neurons (infected with hSyn.2mtYC3.6) (Supplementary Fig. 8A) within 1 h, corroborating the in vivo data. In addition, TgCM decreased the mitochondrial membrane potential (ΔΨm) (measured with TMRE) (Supplementary Fig. 8D), activated the mPTP (evaluated by calcein/cobalt fluorescence) (Supplementary Fig. 8C), and activated caspases 3/7 (detected with CellEvent Caspase 3/7) in primary neurons (Supplementary Fig. 8E). These effects took place in the same time window as the increase of [Ca$^{2+}$]$_{mit}$. WtCM did not drive any of these effects on mitochondria. Blocking of the mPTP with cyclosporine A (CsA)[41], as verified by calcein/cobalt fluorescence, prevented the decrease of ΔΨm, but did not inhibit the increase of [Ca$^{2+}$]$_{mit}$ elicited by TgCM. These results suggest that the reduction of ΔΨm is driven by the increase in mitochondrial Ca$^{2+}$ after TgCM application. Blocking mPTP with CsA showed a trend towards decreasing the number of neurons with caspase 3/7 activation (Supplementary Fig. 8).

**Mitochondrial Ca$^{2+}$ dyshomeostasis in the human AD brain.** Finally, we sought to test whether mitochondrial Ca$^{2+}$ dyshomeostasis is relevant to the human AD brain. Expression of mitochondrial genes encoding proteins involved in Ca$^{2+}$ transport was assessed using gene differential expression data publicly available at the Accelerating Medicines Partnership-Alzheimer's Disease (AMP-AD) Knowledge Portal[42]. We evaluated 25 publicly available microarray and RNA-Sequencing datasets from the Mount Sinai Brain Bank (MSBB)[43] and the Religious Orders Study and Memory and Aging Project (ROSMAP)[44] cohorts, and compared the transcript levels of mitochondrial Ca$^{2+}$-related genes between subjects with AD (Braak stages V–VI) and control individuals (Braak stages 0–I–II), specifically *MCU*, *MCUB*, *MCUR1*, *MICU1*, *MICU2*, *MICU3* and *SMDT1* for mitochondrial Ca$^{2+}$ influx, and *SLC8B1* (encoding NCLX) for mitochondrial Ca$^{2+}$ efflux. Surprisingly, the mitochondrial Ca$^{2+}$ uniporter (*Mcu*) and its regulatory subunits (*Micu1*, *Micu2*, *Micu3* and *Smdt1*) are significantly downregulated ($p < 0.05$ and FDR < 25%) in AD compared to control subjects after adjusting for neuronal loss with the pan-neuronal marker *Map2* (Supplementary Fig. 9, Supplementary Table 1). Interestingly, the Ca$^{2+}$ efflux gene *Slc8b1* is significantly upregulated. *Mcur1* is also downregulated

in the frontal pole and the superior temporal gyrus but fell just below the FDR threshold. No change was observed in *Mcub*. Analyses by CERAD scores (frequent versus absent neuritic amyloid plaques)[45] produced similar results (Supplementary Table 2). All genes for mitochondrial Ca$^{2+}$ influx were downregulated (FDR < 20%), whereas *Slc8b1* was upregulated. *Mcur1* was also upregulated in this case. These results suggest that the expression of most of human genes involved in mitochondrial Ca$^{2+}$ transport is altered in AD patients, denoting a counteracting effect to avoid mitochondrial Ca$^{2+}$ overload.

The present results suggest that Aβ aggregates (likely soluble Aβo) lead to mitochondrial Ca$^{2+}$ overload in AD. Aβ-induced mitochondrial Ca$^{2+}$ uptake requires MCU, as blocking MCU with the specific inhibitor Ru360 prevents the Ca$^{2+}$ overload. In addition, excessive Ca$^{2+}$ taken up by mitochondria leads to opening of the mitochondrial permeability transition pore (mPTP), caspase activation and, eventually, neuronal cell death (Fig. 8).

## Discussion

Proposed mechanisms of neurotoxicity in AD include the disruption of Ca$^{2+}$ homeostasis and mitochondrial dysfunction, which contribute to synaptic and neuronal damage[46]. An extensive array of in vitro and in vivo studies has shown that Aβ aggregates alter cytosolic Ca$^{2+}$ homeostasis. However, prior studies linking Aβ with mitochondrial Ca$^{2+}$ overload[16,17] have used cultured cells and high concentrations of synthetic Aβo, and whether there is mitochondrial Ca$^{2+}$ overload elicited by Aβ aggregates in vivo remained an open question to date. In the present study, we provide evidence of in vivo mitochondrial Ca$^{2+}$ overload in a Tg mouse model of cerebral β-amyloidosis, and after topical application of natural soluble Aβo at concentrations similar to those found in the AD brain.

We found mitochondrial Ca$^{2+}$ overload in the APP/PS1 Tg mouse, a model that recapitulates an accelerated deposition of amyloid plaques. This overload was observed in the Tg mice once the Aβ plaques were present. The mitochondrial Ca$^{2+}$ increase was of a higher magnitude than the cytosolic Ca$^{2+}$ levels observed in the same mouse model[5]. Whereas Ca$^{2+}$ levels in the neuronal cytosol upon exposure to Aβo were in the nM range[7], mitochondrial Ca$^{2+}$ levels were in the μM range. This was not surprising as mitochondria take up large amounts of Ca$^{2+}$ during physiological Ca$^{2+}$ elevations. Indeed, effective mitochondria Ca$^{2+}$ buffering capacity was reported ~0.2–2 μM in isolated brain

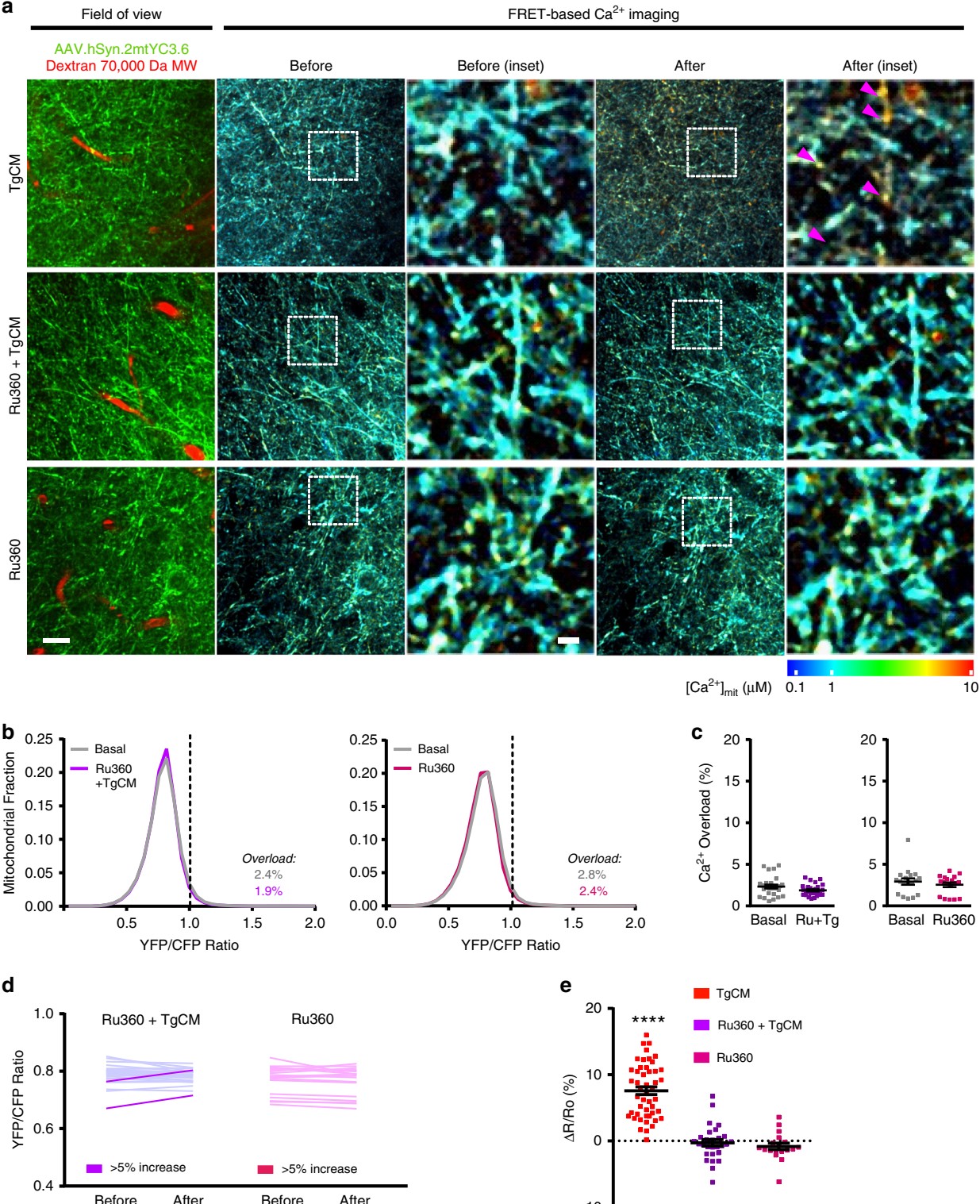

**Fig. 5 MCU is required for the Aβ-induced mitochondrial Ca²⁺ increase in vivo. a** Representative pictures of the effects of the MCU complex blocker Ru360 on the increase of $[Ca^{2+}]_{mit}$ induced by TgCM. Arrow heads show mitochondria with high $[Ca^{2+}]$. Scale bar 20 and 5 μm for the inset. **b** Graphs show histogram of $[Ca^{2+}]_{mit}$ frequency distribution (expressed as YFP/CFP ratio) before and after topical application of TgCM in presence of Ru360 and of Ru360 alone. The percentage of $Ca^{2+}$ overload before (basal) and after TgCM application in presence of Ru360 is noted. **c** Scatter dot plot represents the percentage of mitochondria showing $Ca^{2+}$ overload before and after application of CM for every volume acquired. Error bars represent mean ± SEM (Ru360 + TgCM: before 2.37 ± 0.47% vs. after 1.89 ± 0.27%, non-significant, $n = 28$ z-stacks from 4 mice. Ru360: before 2.82 ± 0.86% vs. after 2.41 ± 0.78%, non-significant, $n = 18$ z-stacks from 3 mice). **d** Averaged YFP/CFP ratios before and after treatment for each z-stack acquired. The dark traces account for the z-stacks that showed an increase ≥5% in YFP/CFP ratio after topical application of the treatment. **e** Scatter dot plot represent the relative change in YFP/CFP ratio for each condition. Bars represent mean ± SEM (7.60 ± 0.81%, 48 z-stacks TgCM; −0.27 ± 1.05%, 28 z-stacks Ru360+TgCM; −0.92 ± 1.25%, 18 z-stacks Ru360 from 7, 4 and 3 mice, respectively, ****$p < 0.0001$).

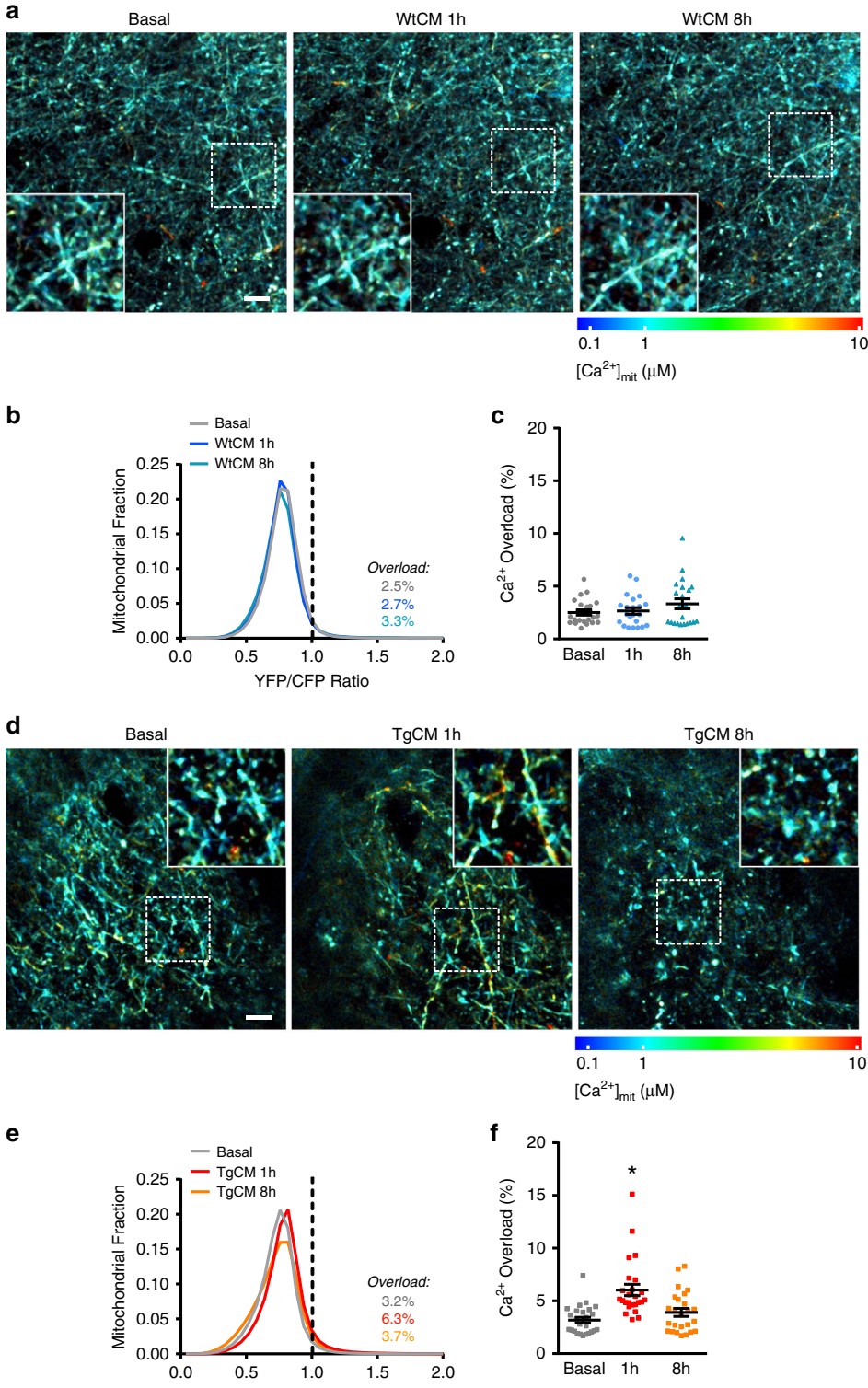

**Fig. 6 Soluble Aβ-driven mitochondrial Ca²⁺ increase recovers after 8 hours post-application. a** Time-lapse images showing mitochondrial [Ca²⁺] increase induced by WtCM, in basal conditions, 1 and 8 h after topical application. Scale bar 20 μm. **b** Histogram of [Ca²⁺]$_{mit}$ frequency distribution (YFP/CFP ratio) before and 1 or 8 h after application of WtCM. The black dotted line corresponds to 2SD above the mean [Ca²⁺]$_{mit}$ measured for all mitochondria before treatment. Ratios above this line were classified as mitochondrial Ca²⁺ overload. **c** Scatter dot plot represents the percentage of mitochondria with Ca²⁺ overload at each time point in every volume acquired. Bars errors represent mean ± SEM (Basal 2.5 ± 0.54%, 1 h 2.7 ± 0.54%; 8 h 3.3 ± 0.54%; non-significant, $n = 22$ z-stacks from 3 mice). **d** Time-lapse images showing mitochondrial [Ca²⁺] increase induced by TgCM, in basal conditions, 1 and 8 h after application. Scale bar 20 μm. **e** Histogram of [Ca²⁺]$_{mit}$ frequency distribution (YFP/CFP ratio) before and 1 or 8 h after application of TgCM. Note that [Ca²⁺]$_{mit}$ distribution is shifted to higher YFP/CFP ratios at 1 h post-TgCM application, but recovers at 8 h post-TgCM application. **f** Scatter dot plot represents the percentage of mitochondria with Ca²⁺ overload at each time point in every volume acquired. Error bars represent mean ± SEM (Basal 3.2 ± 0.60%, 1 h 6.3 ± 1.22%; 8 h 3.7 ± 0.70%; *$p < 0.05$ vs. basal, $n = 25$ z-stacks from 5 mice).

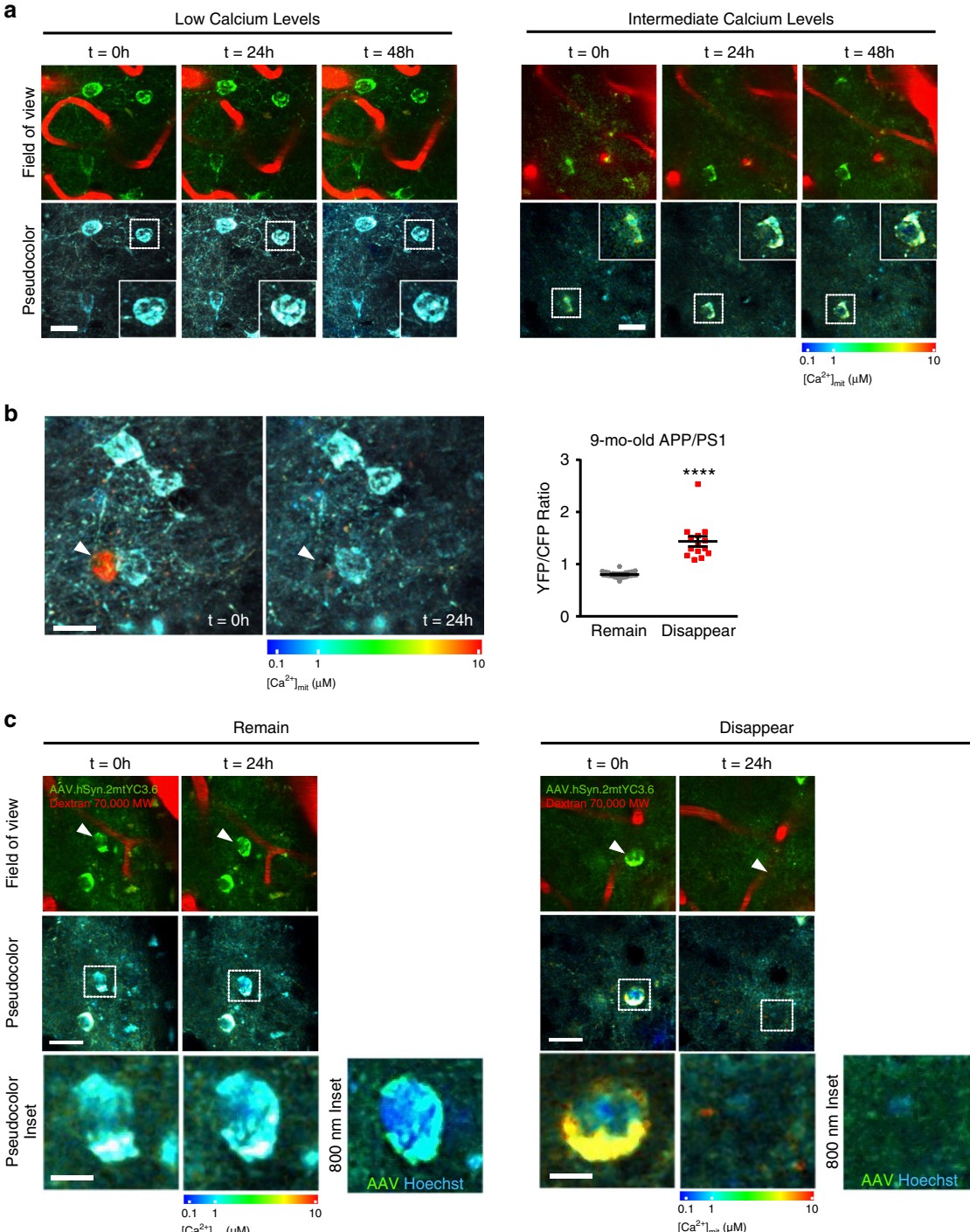

**Fig. 7 Mitochondrial Ca$^{2+}$ overload precedes neuronal death in the APP/PS1 Tg mice. a** Time-lapse images showing that resting mitochondrial Ca$^{2+}$ levels in neurons are steady over time in the 9-mo-old APP/PS1 Tg mouse. The same field of view was imaged at 3 different time points (3 consecutive days). AAV.hSyn.2mtYC3.6 (neurons, green) and Dextran Texas Red (vessels) are shown in the top row and pseudocolor images in the bottom row. Insets show somas of the same neuron during the time lapse. Scale bar 25 μm. **b** Time-lapse images showing mitochondrial Ca$^{2+}$ overload preceding neuronal cell death in a 9-mo-old APP/PS1 Tg mouse. Arrow heads show mitochondrial Ca$^{2+}$ overload in the soma of a neuron, which disappeared 24 h later. Scale bar 20 μm. Scatter dot plot represents YFP/CFP ratio of single neurons present at baseline imaging session that remained or disappeared at 24 h follow-up imaging session as a function of their baseline [Ca$^{2+}$]$_{mit}$. Neurons with a YFP/CFP ratio >1.4 disappeared within 24 h (mean ± SEM, $n = 27$ and 14 neurons from 4 APP/PS1 mice, ****$p < 0.0001$, Student $t$-test). **c** Images show time-lapse recordings of nuclei and mitochondrial Ca$^{2+}$ levels preceding neuronal cell death in a 9-mo-old APP/PS1 Tg mouse. The same field of view was imaged at 2 different time points (2 consecutive days). AAV.hSyn.2mtYC3.6 (neurons, green) and Dextran Texas Red (vessels) are shown in the top row, and pseudocolor images are shown in the middle (scale bar 20 μm) and bottom (scale bar 5 μm) rows. 800 nm inset shows AVV (mitochondria, green) and Hoescht (nuclei, blue) acquired at 800 nm. It was observed that remaining cells (low mitochondrial Ca$^{2+}$) maintained a healthy nucleus (Hoechst), whereas disappearing cells (high mitochondrial Ca$^{2+}$) presented a condensed nucleus.

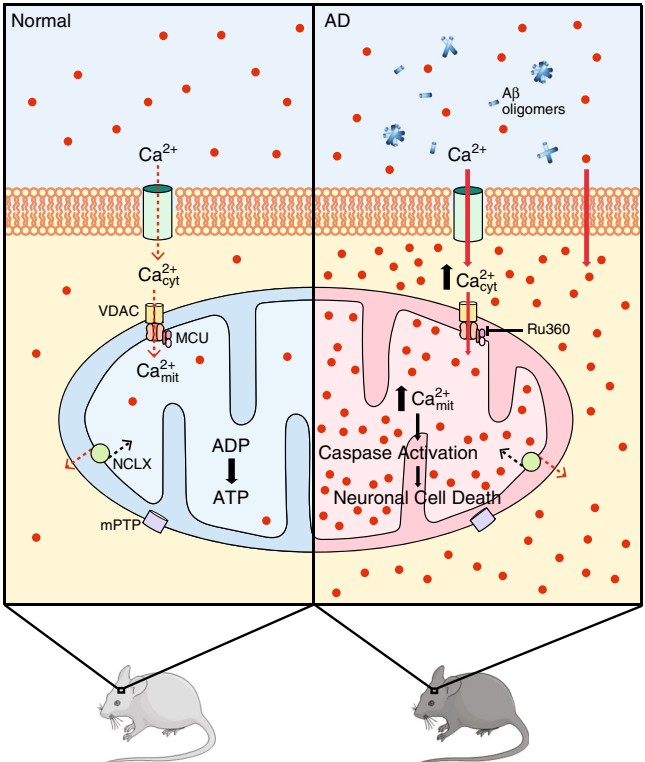

**Fig. 8 Schematic representing the effects of Aβ aggregates on mitochondrial Ca²⁺ homeostasis.** Under physiological conditions (normal), mitochondria take up Ca²⁺ from the cytosol, via mitochondrial Ca²⁺ uniporter (MCU) complex, for activation of mitochondrial metabolism, TCA cycle and OXPHOS, leading to ATP production. However, under pathological conditions (AD), Aβ aggregates (likely soluble Aβo) increase cytosolic Ca²⁺, thus leading to mitochondrial Ca²⁺ overload. Aβ-induced mitochondrial Ca²⁺ uptake is mediated via MCU, as blocking MCU with the specific inhibitor Ru360 prevents it. Excessive Ca²⁺ taken up by mitochondria could lead to opening of the mitochondrial permeability transition pore (mPTP), caspase activation and, eventually, neuronal cell death. This figure was created using modified Servier Medical Art templates, which are licensed under a Creative Commons Attribution 3.0 Unported License (https://creativecommons.org/licenses/by/3.0/).

mitochondria[47] and in neuronal mitochondria in situ[48]. In vivo, we found that mitochondrial Ca²⁺ overload only occurs at levels higher than ~1.6–1.7 μM, comparable to the reported buffering capacity in vitro.

Mitochondrial dysfunction has been suggested as an early event in AD, before plaque deposition[9]. However, several studies have reported no change in resting cytosolic Ca²⁺ levels or neuronal dysfunction before amyloid plaque deposition in AD mouse models[5,24,25]. In agreement with these studies, there was no appreciable mitochondrial Ca²⁺ overload in young mice before plaque deposition (2.5–3-mo-old), suggesting that Aβ deposition is necessary to trigger mitochondrial Ca²⁺ overload in neurons in AD. Further longitudinal studies will be necessary to clarify whether mitochondrial Ca²⁺ elevation occurs concomitantly with plaque formation. Surprisingly, no association was found between mitochondrial Ca²⁺ overload and distance to amyloid plaques, in contrast to cytosolic Ca²⁺ overload, wherein there was a higher likelihood of detecting Ca²⁺ overload in the immediate vicinity of individual plaques[5]. This suggests that diffusible soluble Aβ oligomeric species rather than plaques are likely responsible for the increase of [Ca²⁺]_mit in the neurons of the APP/PS1 mouse, or that mitochondria are less vulnerable than cytosol to the

pathophysiological insult mediated by Aβo. The intrinsic properties of mitochondria with regards to Ca²⁺ kinetics, and as dynamic entities that undergo axonal transport, fusion/fission, and mitophagy when damaged, could also explain this difference.

To confirm that mitochondrial Ca²⁺ overload is triggered by soluble Aβo, we directly applied naturally secreted Aβo (TgCM) to the naive living mouse brain. Application of TgCM to the cortical surface of the mouse brain increased [Ca²⁺]_mit in neurons within 1 h, whereas WtCM or Aβ-immunodepleted TgCM did not. This result supports our previous findings in the APP/PS1 model, and confirms a direct effect of Aβ aggregates altering Ca²⁺ homeostasis in neurons. The reversibility of TgCM-induced mitochondrial Ca²⁺ overload supports a normal extrusion via mitochondrial Ca²⁺ exchangers, and/or clearance of the Aβo from the brain parenchyma.

Mitochondria have been reported as a direct site of Aβ accumulation in neurons in both human AD brains and AD Tg mouse models[49]. However, the mechanism(s) by which Aβ evokes mitochondrial Ca²⁺ uptake in vivo has not been elucidated. Proposed mechanisms include formation of Ca²⁺-permeable pores in the plasma membrane and also in the membranes of intracellular organelles[50], or formation of a non-specific ion channel in the plasma and mitochondrial membranes[51]. On the other hand, Ca²⁺ uptake into the mitochondrial matrix is mediated by the highly conserved MCU complex[29,30], which is selectively blocked by Ru360. Our results show that pre-treatment with Ru360 abolishes TgCM-induced mitochondrial Ca²⁺ uptake, providing evidence that the MCU complex is required for the Aβ-driven mitochondrial Ca²⁺ increase in neurons in the brain in vivo, and arguing against the formation of Ca²⁺-permeable pores or non-specific ion channel in the mitochondrial membrane. However, assessment of protein expression of the mitochondrial-enriched fraction extracted from Wt and APP/PS1 Tg mouse brains did not show any significant differences between groups. Western blot might not be sensitive enough to detect changes in protein levels that affect the small subset of Ca²⁺-overloaded mitochondria seen in vivo. Alternatively, Aβ may alter the levels of a component of the pore or the NCLX exchanger not examined in this model due to lack of adequate antibodies, or may impact their activity rather than expression. Along these lines, recent work by Jadiya et al.[52] has observed a decrease in the expression of NCLX in mitochondria in human AD brains and in mouse models of AD, linked to impaired Ca²⁺ efflux from mitochondria, which could be responsible for the disease progression.

To examine the relevance of mitochondrial Ca²⁺ dyshomeostasis in the human AD brain, we took advantage of publicly available microarray and RNA-Seq datasets, and compared the expression levels of Ca²⁺-related mitochondrial genes in AD patients (Braak stages V–VI) and control subjects (Braak stages 0–I–II). Most of the genes involved in Ca²⁺ influx into mitochondria were significantly downregulated in AD, whereas, the only Ca²⁺ efflux gene was significantly upregulated. These results most certainly point towards a compensatory effect in AD to handle the mitochondrial Ca²⁺ overload, and confirm the existence of a mitochondrial Ca²⁺ dyshomeostasis in the human AD brain.

Optimal mitochondrial activity is crucial to maintain neuronal function and health. Mitochondria buffer Ca²⁺, and in turn, Ca²⁺ modulates mitochondrial activity. Excessive Ca²⁺ accumulation into mitochondria leads to mitochondrial function impairment[53], which could lead to decreased ΔΨm[11] and increased ROS production[40]. Eventually, cytochrome c release and cell death via apoptosis will take place[39]. Remarkably, when APP/PS1 Tg mice were imaged longitudinally, the small fraction of neurons containing mitochondria with elevated [Ca²⁺] in their cell bodies were no longer visible 24 h later, and their nucleus was condensed,

whereas neurons with physiological $[Ca^{2+}]_{mit}$ and a healthy spherical nucleus remained. Surprisingly, whereas the cells with high $[Ca^{2+}]_{mit}$ in the soma were likely to die, sporadic increases of mitochondrial $Ca^{2+}$ within the neurites were not as lethal. This feature suggests that within the soma, high $[Ca^{2+}]_{mit}$ represents an "all or none"-phenomena, and not an isolated event in individual mitochondria. The results obtained here in vivo are supported by previous studies in vitro showing neuronal cell death related to mitochondrial $Ca^{2+}$ overload[16,17], where a subtle or partial depolarization of mitochondria to avoid Aβ-related mitochondrial $Ca^{2+}$ overload in culture cells was protective for neuronal death. Furthermore, our in vitro data show that soluble Aβ increases $[Ca^{2+}]_{mit}$, leading to activation of mPTP, ΔΨm decrease and activation of caspases. Previous studies have already linked Aβ, mPTP and cell death in AD in neurons[54–56] and astrocytes[57,58], but here we show that activation of mPTP is downstream mitochondrial $Ca^{2+}$ overload, as preventing mPTP activation did not forestall mitochondrial Aβo-driven $Ca^{2+}$ overload, but avoided mitochondrial depolarization and caspase activation. Moreover, we have previously shown that there is a rapid cell death once Aβ-triggered oxidative stress invades neuronal cell bodies in vivo in the same mouse model[40]. Oxidative stress started in neurites, but in a small fraction of neurons tracked towards the cell bodies leading to rapid initiation of apoptosis. Mitochondrial $Ca^{2+}$ overload has an important role in the oxidative stress-induced neuronal cell death[59]. Our findings of occasional rapid neuronal loss after mitochondrial $Ca^{2+}$ overload support our previous findings and, more importantly, suggest a close link between high $[Ca^{2+}]_{mit}$, oxidative stress and neuronal apoptosis via mitochondria in AD.

Of note, the neurotoxic effect of Aβo on mitochondria were not restricted to $Ca^{2+}$ elevation, but also involved changes in mitochondrial size and shape. Indeed, the APP/PS1 Tg mouse model exhibits structural abnormalities in mitochondria, including mitochondria swelling and fragmentation[11], and impaired balance of mitochondrial fission and fusion proteins has been shown in brains from AD patients[60]. Aβ-induced mitochondrial fragmentation has been reported to precede neuronal cell death[61]. Further experiments are necessary to monitor the morphological changes and altered dynamics of mitochondria induced by Aβo in vivo.

Here we show that blocking the MCU complex in vivo prevents the Aβ-related mitochondrial $Ca^{2+}$ overload, pointing to a critical role of MCU in the mitochondrial $Ca^{2+}$ overload mediated by Aβo. Multiple lines of evidence support MCU as a candidate therapeutic target against AD: MCU overexpression in mitochondria increased $[Ca^{2+}]_{mit}$ following excitotoxicity through activation of NMDA receptors in vitro, leading to mitochondrial membrane depolarization and cell death[62]. Besides, inhibition of MCU by Ru360, or silencing by siRNA or miRNA, conferred resistance to oxidative stress to the cells in vitro[63]. Also, reducing mitochondrial $Ca^{2+}$ uptake by a null mutation in the gene encoding MCU, improved mitochondrial function and suppressed neurodegeneration in a *C. elegans* model of AD; and preventing mitochondrial $Ca^{2+}$ uptake by pre-treatment with Ru360 decreased ROS levels in familial AD fibroblasts[64]. Our in vitro work shows that preventing activation of the mPTP downstream mitochondrial $Ca^{2+}$ overload could also serve as a therapeutic approach against AD, to protect mitochondria from depolarization (loss of ΔΨm) and subsequent activation of apoptosis.

In summary, our findings demonstrate that Aβ accumulation provokes mitochondrial $Ca^{2+}$ overload in vivo via the MCU complex, leading to neuronal death. We have unveiled a pathophysiological link between $Ca^{2+}$ dyshomeostasis and mitochondrial dysfunction hypotheses of AD pathogenesis and propose the inhibition of the MCU complex and blocking of the mPTP activation as novel therapeutic approaches against AD.

## Methods

**Animals**. Mouse experiments were performed with the approval of the Massachusetts General Hospital Animal Care and under the guidelines of Institutional Animal Care and Use Committees (IACUC, protocol #2004N000047) for the use of experimental animals. Mice used included: (1) APPswe/PSEN1ΔE9 double Tg mice (The Jackson laboratory, B6.Cg-Tg(APPswe,PSEN1dE9)85Dbo/Mmjax, MMRRC Cat# 034832-JAX, RRID:MMRRC_034832-JAX) (APP/PS1, 2- to 3-mo-old and 8- to 10-mo-old) of either sex, expressing both human *APP* gene carrying the Swedish mutation and exon 9 deletion mutation in the *PS1* gene along with age-matched Wt-littermate controls; and (2) C57BL/6J males (4- to 5-mo-old, Charles River) for the application of the conditioned media and Ru360. For preparation of the conditioned media, Tg2576 males (Taconic Farms, B6;SJL-Tg(APPSWE)2576Kha, IMSR Cat# TAC:1349, RRID:IMSR_TAC:1349), which heterozygously overexpress human APPswe under the PrP promoter, were mated with Wt females for preparation of primary cortical neurons. Mice were socially housed at 3–4 animals per cage with ad libitum access to food and water on a 12/12 h light/dark cycle with controlled conditions of temperature and humidity.

**hsyn.2mtYC3.6 and CBA.YC3.6 AAV construction and production**. The expression of the $Ca^{2+}$ indicator Yellow Cameleon 3.6 (YC3.6)[19] was targeted to mitochondria by using the mitochondrial targeting sequence cytochrome oxidase subunit 8 A (COX8A) at the N terminus, duplicated in tandem to enhance the specificity of its localization[65]. 2mtYC3.6 was inserted on the 5'-end and in frame with the cDNA of YC3.6, which we had previously cloned in an AAV.CBA.2mtYC3.6-WPRE (woodchuck hepatitis virus post-transcriptional regulatory element) backbone. We then cut out the 2mtYC3.6 cDNA (*Nhe*I/*Xho*I-fill-in) and subcloned it into an AAV-hSyn backbone obtained from the UPENN vector core, obtaining the final AAV.hSyn.2mtYC3.6-WPRE plasmid. The integrity of the inverted terminal repeats (ITRs) was verified by sequencing and SmaI digestion. Flanked by the ITRs, the expression cassette included the following components, (1) a 1.7-kb sequence containing human synapsin 1 gene promoter, (2) 2mt (mitochondrial targeting sequence), (3) YC3.6, (4) WPRE and (5) Simian virus 40 (SV40). See construct in Fig. 1a. High titers of AAV serotype 8 were produced using triple transfection in HEK293 cells by the Schepens Eye Research Institute Gene Transfer Vector Core. The virus titer was $5.7 \times 10^{12}$ viral genomes copies per mL.

To measure cytosolic $Ca^{2+}$ concentration we used the construct pAAV.CBA. YC3.6-WPRE[5]. The plasmid contained AAV terminal repeats (ITRs) and the expression cassette, which included the following components: the hybrid cytomegalovirus (CMV) immediate-early enhancer/chicken β-actin promoter/exon1/intron; yellow cameleon 3.6 cDNA and woodchuck hepatitis virus post-transcriptional regulatory element (WPRE). High titers of AAV serotype 8 were produced using triple transfection in HEK293 cells by the Schepens Eye Research Institute Gene Transfer Vector Core. The virus titer was $1.1 \times 10^{12}$ viral genomes copies per mL.

**Cell culture and transfection**. Primary neuronal cultures were obtained from cerebral cortex of CD1 mouse embryos (Charles Rives Laboratories) at gestation day 14–16. Neurons were dissociated using Papain dissociation system (Worthington Biochemical Corporation, Lakewood, NJ, USA) and were maintained for 10–14 days in vitro (DIV) in Neurobasal medium containing 2% B27 supplement (Gibco), 1% penicillin/streptomycin (Gibco) and 1% glutamax (Gibco) in a humidified 37 °C incubator with 5% $CO_2$ without further media exchange[5].

N2a cells were maintained in OptiMEM (Gibco) supplemented with 5% Fetal Bovine Serum (FBS) (Atlanta Biologics) and 1% penicillin and 1% streptomycin (Gibco) in a humidified 37 °C incubator with 5% $CO_2$.

Cells were transiently transfected at 70–80% confluence with 4 μg of hSyn.2mtYC3.6 and mRuby-Mito-7 (gift from Michael Davidson Addgene, #55874) plasmids using Lipofectamine 2000 (Life Technologies) according to the manufacturer's instructions.

**In situ calibration of hSyn.2mtYC3.6 in N2a cells**. N2a cells were plated in 35 mm glass-bottom dishes (MatTek Corporation) and transfected with lipofectamine 2000 (Life Technologies) and 4 μg of the hSyn.2mtYC3.6 construct. 24 h after transfection, cells were permeabilized for 1 min with 100 μM digitonin in intracellular medium (130 mM KCl, 5 mM NaCl, 1 mM $MgCl_2$, 2 mM $KH_2PO_4$, 20 mM HEPES, 0.5 mM EGTA, 5 mM succinate, 1 mM ATP, pH 7), followed by three washes with intracellular medium devoid of digitonin. After permeabilization, cells were exposed to solutions containing known $Ca^{2+}$ concentration (ranging from 0 μM to 1 mM) and 10 μM FCCP, to depolarize mitochondria and equilibrate the $[Ca^{2+}]$ in the medium and in the organelle[66]. Calibration solutions with free $Ca^{2+}$ concentration <20 μM were prepared by mixing intracellular media with the chelators EGTA or HEDTA and 1 M $CaCl_2$ according to MaxChelator Ca-EGTA Calculator (https://somapp.ucdmc.ucdavis.edu/pharmacology/bers/maxchelator/CaEGTA-NIST.htm) and to Park and Palmer[67]. Calibration solutions with free

$Ca^{2+}$ concentration >20 μM were made by adding various volumes of 1 M $CaCl_2$ to the intracellular media solution. 2 photon excitation was used at 860 nm, and emitted light was collected in three channels in the range of 380–480, 500–540 and 560–650 nm. Images were acquired on an Olympus FluoView FV1000MPE multiphoton laser-scanning system mounted on an Olympus Bx61WI microscope, and photomultiplier settings were selected and maintained for the intravital imaging. YFP/CFP fluorescence ratios were plotted against free $Ca^{2+}$ concentration and fitted with a sigmoid equation. The apparent $K_d$ ($K_d'$) and Hill coefficient ($n$) were determined as described previously[20]. Custom-written ImageJ software (https://imagej.nih.gov/ij/) was used for analysis and calculation of YFP/CFP ratios. Note that ratios increased with increasing [$Ca^{2+}$]. Ratios were then converted to free $Ca^{2+}$ concentration with standard ratiometric equations[20]:

$$[Ca^{2+}]_{mit} = K_d' \left( \frac{R - R_{min}}{R_{max} - R} \right)^{1/n}$$

**Preparation of Wt and TgCM and Aβ-immunodepleted media.** Primary neuronal cultures were prepared as described above[5]. Neurons were plated on 35-mm culture PDL-coated dishes. Tissue from each embryo was collected for genotyping by PCR. Neurons were maintained in Neurobasal media (Gibco) containing 2% B27 supplement (Gibco), 1% penicillin/streptomycin (Gibco) and 1% glutamax (Gibco) for 14 DIV. Conditioned media from either Tg cultures (TgCM) or their Wt littermates (WtCM) was collected, and the concentration of $Aβ_{40}$ was measured by mouse/human ELISA kit (Wako #294-64701 Human/Rat $Aβ_{1-40}$).

Immunodepletion of human Aβ from the TgCM was performed overnight by using the mouse antibody 6E10 (Purified anti-β-Amyloid, 1–16 antibody BioLegend Cat# 803004, RRID: AB_2715854) and protein G sepharose beads (Sigma-Aldrich). Briefly, protein G beads were washed with cold Neurobasal medium in order to avoid non-specific binding. Then, 1 mL of TgCM was incubated with 40 μL of the pre-washed protein G beads and 6 μg of 6E10 antibody overnight at 4 °C under rotation. Finally, the supernatant was collected and the Aβ concentration was quantified by mouse/human ELISA kit (Wako #294-64701 Human/Rat $Aβ_{1-40}$) according to the manufacturer's instructions.

**Mitochondrial and cytosolic calcium in vitro imaging.** Primary cortical neurons obtained from CD1 embryos were cultured for 12–15 days in vitro (DIV) and infected with AAV.hSyn.2mtYC3.6 to target mitochondria. Experiments were performed using sister cultures obtained on the same day to control for any culture-to-culture variation. Three days after infection, cells were incubated with fresh conditioned media and subjected to multiphoton microscopy. 5–6 fields of view were imaged randomly. For some experiments cortical neurons infected with the mitochondrial reporter hSyn.2mtYC3.6 were loaded with the calcium indicator calbryte 590 (0.5 μM) for 30 min at room temperature to measure $Ca^{2+}$ changes in the cytosol and mitochondria simultaneously. Images were acquired in an Olympus FluoView FV1000MPE multiphoton laser-scanning system mounted on an Olympus Bx61WI microscope. Two photon excitation was used at 800 nm (for calbryte) and 860 nm (for 2mtYC3.6), and emitted light was collected in the range of 380–480, 500–540 and 560–650 nm. Either images were acquired in basal conditions and after treatment with conditioned media, or time-lapse recordings were acquired following stimulation with KCl (50 mM) or TgCM. Background fluorescence corresponding to regions devoid of cells was subtracted. Images were analyzed using custom-written MATLAB scripts (see Image Analysis section).

**Mitochondrial permeability transition pore (mPTP) activity.** mPTP opening was directly assessed by the calcein/cobalt method[68]. Briefly, primary cortical neurons cultured for 12–15 DIV were co-loaded with calcein-AM 1 μM, $CoCl_2$ 1 mM and Hoechst 33342 1 μM for 15 min at room temperature and imaged with confocal microscopy. The effect of TgCM was evaluated longitudinally. Images were acquired in the green channel (488 nm) every 10 min. Cells were maintained at 37 °C and 5% $CO_2$ during the recording. Fluorescence traces from mitochondria in individual cells were normalized relative to their value before addition of any solutions ($t = 0$ min). Background fluorescence corresponding to regions devoid of cells was subtracted. In some experiments, cells were pre-incubated with cyclosporine A 1 μM for 15 min before and co-applied during the recording.

**Mitochondrial membrane potential (ΔΨm) imaging.** The effects of TgCM on ΔΨm were evaluated by confocal microscopy in cells loaded with the ΔΨm sensitive probe TMRE. Primary cortical neurons cultured for 12–15 DIV were loaded with TMRE (10 nM) and Hoechst 33342 1 μM for 15 min at room temperature and imaged with confocal microscopy. The effect of TgCM was evaluated longitudinally. Cells were maintained at 37 °C and 5% $CO_2$ during the recording. Excitation of the neurons was performed at 361 nm for Hoechst and 549 nm for TMRE, and images were taken every 10 min. Fluorescence traces from individual cells were normalized relative to their value before addition of any solutions ($t = 0$ min). Background fluorescence corresponding to regions devoid of cells was subtracted. In some experiments, cells were pre-incubated with cyclosporine A 1 μM for 15 min before and co-applied during the recording.

**Caspase activation imaging.** Caspase activation was detected with CellEvent Caspase 3/7 Green Detection Reagent in primary cortical neurons at 12–15 DIV. Neurons were labeled with CellEvent Caspase 3/7 5 μM and Hoechst 33342 1 μM for 15 min at room temperature. Then media was replaced by the conditioned media and green fluorescence was detected under confocal microscopy. Excitation of the neurons was performed at 361 nm for Hoechst and 488 nm for CellEvent, and images were recorded every 10 min. Cells were maintained at 37 °C and 5% $CO_2$ during the recording. Fluorescence traces from individual cells were normalized relative to their value before addition of any solutions ($t = 0$ min). Background fluorescence corresponding to regions devoid of cells was subtracted. In some experiments, cells were pre-incubated with cyclosporine A 1 μM for 15 min before and co-applied during the recording.

**Stereotactic intracortical injection of AAVs.** Four- to five-mo-old C57BL/6 J Wt mice were injected intracortically with AAV.hSyn.2mtYC3.6 (serotype 2/8) or pAAV.CBA.YC3.6-WPRE (serotype 2/8). Stereotactic intracortical injections were performed as previously described[7]. Briefly, mice were anesthetized with 1.5% (vol/vol) isoflurane and positioned on a stereotactic frame (Kopf Instruments). Body temperature was maintained throughout the surgery with a heating pad. 3 μL of viral suspension was injected in the somatosensory cortex using a 33-gauge sharp needle (Hamilton Medical) attached to a 10 μL Hamilton microsyringe (Hamilton Medical), at a rate of 0.15 μL/min. Stereotactic coordinates of the injection sites were calculated from bregma (anteroposterior −1 mm, mediolateral ±1 mm and dorsoventral −0.8 mm).

Eight- to nine-mo-old APP/PS1 Tg mice were injected intracortically with AAV.hSyn.2mtYC3.6 at the moment of the cranial window implantation with 3 μL of viral suspension in the somatosensory cortex (left hemisphere) at 0.15 μL/min, followed by cranial window implantation. In some cases, 1 μl of Hoechst 33342 1uM was co-injected with AAV.hSyn.2mtYC3.6 to label nuclei in the area of injection. Mice were given buprenorphine (0.1 mg/kg) for 3 days following surgery.

**Cranial window implantation.** In the acute conditioned media experiments, a 6 mm cranial window was implanted in the C57BL/6J Wt mice three to four weeks after AAV.hSyn.2mtYC3.6 or pAAV.CBA.YC3.6-WPRE injection. Mice were anesthetized with 1.5% (vol/vol) isoflurane and placed in a stereotactic apparatus. A piece of skull over somatosensory cortex was removed and replaced with an 8 mm diameter glass coverslip[7]. Dura matter was removed before window installation and the window was sealed containing PBS. Texas Red dextran (70,000 MW; 12.5 mg/mL in PBS; Molecular Probes) was retro-orbitally injected to provide a fluorescent angiogram. An imaging session was first performed to determine the basal resting [$Ca^{2+}$]$_{mit}$. Then, the window was removed and sealed again with WtCM, TgCM, Aβ-immunodepleted TgCM, Ru360 (Calbiochem, Merck Millipore) or Ru360 + TgCM (final volume applied was 40 μL). In the last case, Ru360 was pre-incubated for 15 min previous application of TgCM. After 1 h, each mouse was reimaged at the same fields of view to determine the relative changes in [$Ca^{2+}$] ($\Delta R/R_0$).

For the APP/PS1 Tg mice, cranial windows were implanted to the animals following the intracortical virus injection. Mice were anesthetized with 1.5% (vol/vol) isoflurane and placed in a stereotactic apparatus. A piece of skull over left somatosensory cortex was removed, replaced with a 5 mm diameter glass coverslip and fixed with Krazy Glue and dental cement[7]. Mice were given buprenorphine (0.1 mg/kg) for 3 days following surgery. Expression of the virus was allowed for 3–4 weeks before imaging. Methoxy-XO4 (4 mg/kg) was intraperitoneally injected 1 day before the imaging session to label amyloid plaques[21].

**In vivo multiphoton microscopy imaging.** Mice were anesthetized with isoflurane (5% inhalation, mixed with pure $O_2$) for induction, and a reduced concentration of isoflurane (~1.5%) was used during the imaging. The animal's body temperature was maintained at ~37.5 °C during the imaging session with a heating pad; ophthalmic ointment was applied to protect the eyes. Mice were visually monitored throughout the duration of the imaging session (which lasted on average 2 h). Images of AAV.hSyn.2mtYC3.6 or pAAV.CBA.YC3.6-WPRE expressing neurons, amyloid pathology and Texas Dextran Red angiograms were obtained by using an Olympus FlouView FV1000MPE multiphoton laser-scanning system mounted on an Olympus Bx61WI microscope and an Olympus 25× dipping objective (NA = 1.05). A Deep-See Mai Tai Ti:Sapphire mode-locked laser (Mai Tai; Spectra-physics) generated multiphoton excitation at 860 nm, and three photomultiplier tubes (PMTs) (Hamamatsu) collected the emitted light in the range of 380–480, 500–540 and 560–650 nm[69]. Images were taken at 5× zoom. Five to fifteen cortical volumes (Z-series, 127 μm × 127 μm, 200–300 μm depth) were acquired per mouse, at a resolution of 512 × 512 pixels. CFP and YFP PMTs settings remained unchanged throughout the different imaging sessions. Laser power was adjusted as needed and kept below 50 mW to avoid phototoxicity[70].

At the end of the last imaging session, APP/PS1 Tg mice were euthanized under $CO_2$, perfused with phosphate buffered saline (PBS, Gibco) and brains were removed. The hemisphere injected with AAV.hSyn.2mtYC3.6 was fixed with 4% paraformaldehyde and 15% glycerol for 24 h, frozen in mounting media on ice cold isopropanol and sliced into 20 μm sections on a cryostat (Leica). The other hemisphere was flash frozen with liquid nitrogen and stored at −80 °C until use.

**Image analysis**. Multiphoton microscopy imaging data, acquired in vivo, were analyzed by using a custom-written MATLAB (MathWorks) script. To perform automatic segmentation of mitochondria, an adaptive thresholding procedure was applied to generate binary images and individual mitochondria were selected using a constraint on object size. Amyloid plaques were manually extracted from the $z$-stacks. To obtain Yellow Cameleon ratio values, the data were preprocessed by subtracting the background from the CFP and YFP channels (mean of the bottom 5% of intensity values in each slice of the $z$-stack) and applying a mean filter with radius 2 to each channel. The ratio for each mitochondrion was calculated by dividing the sum of the intensity values corresponding to the mitochondrion in the YFP channel by the sum of intensity values in the CFP channel. The YFP/CFP ratios were mapped to mitochondrial $Ca^{2+}$ concentrations using the in situ calibration curves (see In situ calibration of 2mtYC3.6 in N2a cells section). Calcium pseudocolored images were created using MATLAB by first assigning the $[Ca^{2+}]$ values to a RGB (red, green, blue) colormap with the color range determined by the empirical values for $R_{min}$ and $R_{max}$. The RGB image was then converted to an HSV (hue, saturation, value) image with the 'value' field set to the mean of the YFP and CFP intensity images. Distance to plaque was measured based on the centroid of the mitochondrion to the edge of the nearest plaque. To assess changes in mitochondrial morphology, a unique slice per $z$-stack was selected, matching the region across the three different time points (basal, 1 h, 8 h). Slices were selected within 30-60 μm depth. ROIs were drawn around individual mitochondria with the free hand selection tool in ImageJ based on the pseudocolor images. About 100–125 random individual mitochondria were taken. Matlab was used to measure the parameters of interest (area, perimeter and circularity) for each ROI. Area and perimeter were determined as the total number of pixels within the ROI and the number of pixels outlining the ROI, respectively, and then converted to physical units of μm$^2$ and μm. Circularity was calculated using the formula, Perimeter$^2$/4πArea. Images presented in the figures are either a single slice from the $z$-stack or 2D maximum intensity projections of the $z$-stack.

**Mitochondrial-enriched fraction and Western blotting**. Frozen hemibrains devoid of cerebellum from APP/PS1 mice were lysed in Tris-buffered saline (TBS) with 2% sodium dodecyl sulfate (SDS) and protease inhibitor (Roche Diagnostics) at 10% (w/vol) in a 5 mL glass dounce homogenizer. Tissue was dounce homogenized with 25 up/down strokes by hand and incubated for 30 min at 37 °C. The homogenate was then centrifuged at 100,000 × g for 30 min at 20 °C. The supernatant was collected, aliquoted and stored at −80 °C until use. Mitochondrial-enriched fraction was obtained using a mitochondrial isolation kit (mitochondrial isolation kit for mammalian tissue samples, Abcam) according to the manufacturer's instructions. A bicinchonic acid assay (BCA, Thermo Scientific Pierce) was used to determine protein sample concentration. Supernatants were subjected to SDS-PAGE on NuPAGE Bis-tris gels (Life Technologies). Briefly, 20 μg of total protein per well were loaded on 4–12% Bis-Tris SDS-PAGE gels and run in MES buffer (Invitrogen). Proteins were then transferred to PVDF membranes (Millipore), which were incubated overnight with the primary antibodies: rabbit anti-MCU (1:1000, Sigma-Aldrich Cat# HPA016480, RRID:AB_2071893), rabbit anti-MiCU1 (1:1000, Sigma-Aldrich Cat# HPA037480, RRID:AB_10696934), rabbit anti-EFHA1 (MiCU2) (1:500, Abcam Cat# ab101465, RRID:AB_10711219), rabbit anti-SLC24A6 (NCLX) (1:250, Sigma-Aldrich Cat# SAB2102181, RRID:AB_10606696), mouse anti-VDAC1/porin (1:1000, Abcam Cat# ab14734, RRID:AB_443084), mouse anti-β-tubulin I (1: 10,000, Sigma-Aldrich Cat# T7816, RRID:AB_261770). The next day, blots were incubated with infrared secondary goat anti-mouse IgG IRDye680 or goat anti-rabbit IgG IRDye800 (1: 5,000, LICOR) for 1 h at room temperature, and imaged on an Odyssey Infrared Imaging System (LICOR). Blots were converted to gray scale and densitometry analysis was performed using Image Studio Lite Ver5.2.

**Immunohistochemistry**. 20 μm coronal sections were subjected to antigen retrieval, permeabilized with 0.5% triton X-100 and incubated with antibodies against GFP (chicken anti-GFP IgY, 1:500, Aves Labs Cat# GFP-1020, RRID: AB_2307313), HSP60 (rabbit anti-HSP60, 1:500, Abcam Cat# ab46798, RRID:AB_881444), NeuN (mouse anti-NeuN, 1:500 Millipore Cat# MAB377, RRID:AB_2298772) and GS (rabbit anti-Glutamine Synthetase, 1:500, Abcam Cat# ab73593, RRID: AB_2247588) overnight at 4 °C. Appropriate secondary antibodies (Alexa fluor 488-conjugated goat anti-chicken IgG antibody, 1:500, Molecular Probes Cat# A-11039, RRID:AB_142924, Cy3-conjugated goat anti-rabbit IgG antibody, 1:1000, Abcam Cat# ab6939, RRID:AB_955021, Cy3-conjugated goat anti-mouse IgG antibody, 1:1000, Abcam Cat# ab97035, RRID:AB_10680176, and Cy5-conjugated goat anti-rabbit IgG antibody, 1:1000, Jackson ImmunoResearch Labs Cat# 111-175-144, RRID:AB_2338013) were applied and incubated for 1 h at room temperature. Slices were mounted with DAPI Vectashield (Vector Laboratories) and subjected to fluorescence imaging.

**Analysis of publicly available human brain datasets**. 25 microarray and RNA-Sequencing (RNA-seq) datasets from the Mount Sinai Brain Bank (MSBB)[43] and the Religious Orders Study and Memory and Aging Project (ROSMAP)[44] cohorts, spanning 19 brain regions, were analyzed as described in Bihlmeyer et al.[71] Four of these brain regions had both microarray and RNA-Seq data. The samples were grouped by the Braak score, which assesses the distribution of tau neurofibrillary

tangles in the subject's brain[72]. The expression of 8 mitochondrial genes involved in mitochondrial $Ca^{2+}$ homeostasis (*MCU, MCUB, MCUR1, MICU1, MICU2, MICU3, SMDT1* and *SLC8B1*) was compared in AD Braak stages (V–VI) versus control subjects (Braak 0–I–II) and by CERAD analysis (frequent versus absent neuritic amyloid plaques). Data were adjusted for neuronal loss with the pan-neuronal marker MAP2. Multiple comparisons corrections were performed using the Benjamini–Hochberg method[73]. For the purpose of selection for visual display, a FDR threshold of 0.25 was used. All results for the differential expression analysis are listed in Supplementary Tables S1 and S2.

**Statistics**. Statistical analyses were conducted in SAS and GraphPad 5.0. Data are reported as mean ± SEM. To assess differences between groups (Wt vs. APP/PS1; Basal vs. WtCM/TgCM/Aβ-depleted TgCM/TgCM+Ru360/Ru360), global and pair-wise, for all measurements of interest (YFP/CFP Ratio, $Ca^{2+}$ overload and $\Delta R/R_0$), a linear mixed effects model was fitted with treatment group as fixed effect and mouse as random effect. The least square mean and standard error estimates and the $p$-values resulting from these models are presented in the figure legends. Similarly, estimates for basal, 1 h post CM application and 8 h post CM application mean and SE measurements were generated from a linear mixed effects model with a fixed effect of an intercept only and a random effect of mouse. Kruskal–Wallis test was used for the analysis of the relationship between percent of $Ca^{2+}$ overload and distance to the edge of the plaque. $p < 0.05$ was considered significant.

**Reporting summary**. Further information on research design is available in the Nature Research Reporting Summary linked to this article.

## Data availability
The authors declare that all data supporting the findings of this study are available within the article and its Supplementary Information files or from the corresponding author upon reasonable request. The source data underlying Figs. 1e, 2c–f, 3b, 4c–g, 5b–e, 6b, c, e, f, 7b, Supplementary Figs. 2A, B, 3B–E, 4B–G, 5B, 7A–F, 8A, C–E can be found in the Source Data file. The plasmids will be made available from the corresponding author upon reasonable request.

## Code availability
Customized MATLAB scripts generated for this study are available from the corresponding author on request.

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

## Acknowledgements

This work was supported by NIHR01AG0442603, S10 RR025645 and R56AG060974 (BJB); and by the Tosteson & Fund for Medical Discovery and the BrightFocus Foundation A2019488F (MCR). MSBB study was supported by the grants AG046170, AG054014, AG057440 and AG057907 from the NIH/National Institute on Aging (NIA). A.S.-P. was supported by the Alzheimer's Association (AACF-17-524184) and the National Institute for Neurodegenerative Diseases and Stroke (NINDS R25NS065743). The ROSMAP project was supported by funding from the National Institute on Aging (AG034504 and AG041232). We acknowledge the Schepens Eye Research Institute Gene Transfer Vector Core for the AAV.hSyn.2mtYC3.6 and AAV.CBA.YC3.6.WPRE vector production.

## Author contributions

M.C.-R. designed experiments, collected and analyzed data and wrote the original draft. S.S.H. wrote ImageJ and Matlab macros and edited the manuscript. A.C.S and E.K.K. helped with immunohistochemistry and mouse brain injections. S.D. analyzed microarray and RNA-seq data. Z.F. performed cloning design. A.N.R. helped preparing conditioned media and primary neurons. A.M. performed the statistical analyses. M.G.A. helped with mouse brain injections. A.S.P. provided expertise and feedback and edited the manuscript. E.H. performed cloning design, provided expertise and feedback and edited the manuscript. B.J.B. conceptualized the research, designed experiments, discussed data, edited the manuscript and secured funding. All authors read and approved the final manuscript.

## Competing interests

The authors declare no competing interests.
