## [Peer Review File · Nature Communications]

Reviewers' Comments:

Reviewer #1:

Remarks to the Author:

This manuscript investigates the calcium handling of neuronal mitochondria in amyloid-driven mouse models of Alzheimer's disease (AD). There is a well-documented mitochondrial pathology associated with AD and it has already been shown in neuronal cultures that application of amyloid or expression of disease-associated variants of the amyloid pathway can lead to abnormal calcium handling in mitochondria. This study extends those findings to an in vivo context using a virally-delivered fluorescent calcium reporter to neuronal mitochondria in the cortex of mice – multiphoton microscopy was used to visualize the calcium sensor. The approach is an attractive one and, of course, it is important to know how and when mitochondrial biology goes awry in AD pathology. However, in this study, the major finding is that mitochondrial calcium handling is aberrant in an amyloid-dependent manner. This largely recapitulates what has been demonstrated in vitro, albeit in the living brain. There are some attempts to address how mitochondrial calcium goes wrong and what the consequences might be, but these mechanisms are not fully explored. For example, what links amyloid to mitochondrial calcium handling? what are the consequences of stopping calcium overload for pathology in the transgenic mice? Furthermore, there are a substantial number of specific concerns (detailed below) about methods and interpretation that need addressing. These uncertainties limit my enthusiasm for publication of the manuscript as it is.

Specific points:

Figure 1C is not at all convincing. There is so much COX-IV staining (naturally) that it would be difficult to rule out false colocalization, yet, in fact, the image makes it seem that there are locations with YC3.6 fluorescence (green) that are not COX-IV (red) positive. Perhaps some quantification, or a different imaging approach would help.

It is argued that the mitochondrial environment may change the properties of the YC3.6 - I'd agree that this could well be the case given the sensitivity of fluorescent proteins to redox state, pH and oxygen availability. However, these properties could equally be affected in AD neurons. So, how do we know that the change in FRET ratio is not an effect of differential mitochondrial environment in AD mice, potentially independent of calcium. A calibration of YC3.6 in AD neurons would confirm the way it performs.

It is not clear in the methods if (and how) mice were anaesthetized for imaging – details are required. How is anesthesia depth monitored? is it the same in WT and APP mice? this is crucial as we know anaesthesia depth will affect neuronal activity and that activity will influence calcium in the cells and mitochondria.

Figure 2E – the definition of Ca overload appears to be largely arbitrary. And the analysis of it is a simple recapitulation of the data in Figure 2C & 2D. It only really makes sense to define Ca overload if that level relates to some kind of mitochondrial pathology, irreversibility of the Ca overload or perhaps neuronal activity.

Figure 2D - The assertion that "only a small fraction of the total number of mitochondria exhibited elevated Ca²⁺" appears to be at odds with the very similarly-shaped distributions. Indeed, the curves in Figure 2D suggests that there is a general shift in Ca across the population.

In many AD mouse models, neuronal activity has been shown to be perturbed. If neurons are firing differently then could the change in mitochondrial calcium taken as a snapshot not simply reflect that?

Is the calcium overload in specific neurons or mitochondria (e.g. Figure 2B) persistent over time (i.e. can it be seen over multiple imaging sessions?)?

For topical application of A β (Figure 4), it is not clear why young (4-5 months) mice were used compared to the effects on older (9 months) mice in transgenic model. Please justify and explain if this makes a difference when comparing experimental regimes.

The block of A β -induced increase in mitochondrial calcium by Ru360 does suggest that the MCU is involved. However, it is possible that Ru360 renders neurons completely insensitive to A β , perhaps by altering their activity (and plasma membrane calcium entry pathways), or even damaging/killing the cells. It would be good to know that the neurons pre-treated with Ru360 are still functional in some sense.

Figure 6 and Supplementary Figure 3 - The mitochondrial morphology analysis following A β application is underdone. It is not clear from the images shown how shape would actually be determined given the density of staining and whether the data in Supplementary Figure 3 relates specifically to mitochondria showing calcium overload (this would be a prediction perhaps?). Apparent changes in size could be manifest from small structures like mitochondria purely due to an increase in the fluorescence signal (were both CFP and YFP channels used in the size analysis?). Were similar morphological changes seen in the transgenic mice? Neuronal mitochondria have very different structure in axons and dendrites - can the morphology data be related to any particular type/location?

Figure 7 - it is important to establish that this neuronal death effect is specific to the AD transgenic mice (as opposed to a non-specific viral or protein expression effect). As such, the WT control is an important omission.

The relevance of the human data analysis to the rest of the study is loose at best. We already know that AD is associated with mitochondrial defects and these analyses confirm there are changes. However, without further experiments to link the human data to the mouse findings, Figure 8 should not be part of this manuscript.

Line 72 - "YC3.6 is sufficiently bright and exhibits a large dynamic range, resulting in an excellent signal-to-noise ratio." - This needs to be more specific ("sufficiently bright" for what?) and needs to refer to published data or data contained in this manuscript.

Reviewer #2:

Remarks to the Author:

Thank you for asking to me to read this interesting paper. The authors have used a genetically encoded calcium sensitive reporter targeted to neuronal mitochondria to explore changes in mitochondrial calcium content and signalling in a transgenic mouse overexpressing Abeta, show that there is an increased mitochondrial calcium concentration in neurons close to amyloid plaques, and argue that this might drive progression to cell death. In the latter part of the work the authors have explored the expression levels of protein components of the mitochondrial calcium influx and efflux pathways and - rather oddly, perhaps - there is no change in the mouse but that show that, in genetic databases from human material, the components of the MCU complex are downregulated while the NLCX is upregulated

The concept is not novel but the experimental data in an in vivo model is potentially very interesting.

The work is nicely presented, and seems to have been well done as one would expect from this

excellent lab., and follows on from a number of papers from this lab along a rather similar theme. I do need to raise some issues however.

i) In order to interpret the meaning of a rise in mitochondrial calcium content it is critical to know what is happening to cytosolic calcium signals. The mitochondrial signal follows the cytosolic signal - if the cytosolic calcium signal is elevated it is not surprising if the mitochondrial signal is also elevated. If the cytosolic signal is normal, then a rise in the mitochondrial signal appears as a more specific phenomenon driving the pathophysiology. It is very hard to judge whether this relationship has changed in any way as the only comparator is an old paper from the same group that showed (I believe) a very modest increase in cytosolic calcium content in neurons close to the plaques. To understand this better, it would be best to look at dynamic changes in cytosolic and mitochondrial calcium in the same cells if that is possible and ask whether this changes in the disease models.

ii) Showing a rise in mitochondrial calcium in response to 300mM (?) KCl used as shown doesn't really tell us very much except that the probe works. At the very least I am surprised that the authors haven't asked how this response changes in the disease model.

iii) 300mM KCl will surely be very hypertonic?

iv) Previous work from this group suggested that mitochondrial membrane potential is selectively reduced in neurons close to plaques. They have also shown an absence of mitochondria close to plaques which makes the present work even more complex to interpret. As mitochondrial calcium uptake is potential dependent, the observations become rather surprising and need explanation. What is driving a rise in mitochondrial calcium signal despite a reduced potential? Or is the reduced potential driven by the increased mitochondrial calcium content and PTP opening? This is experimentally testable either by chelating calcium or by inhibiting the PTP.

v) I understand that the calcium reporter should be specific for neurons (this should be stated more clearly) but perhaps some immunofluorescence should confirm that it is selectively expressed in neurons?

vi) The final scheme appears to suggest that Abeta promotes increased calcium through voltage gated calcium channels - I can't see here any evidence to support that model.

vii) The final scheme also suggests that the rise in mitochondrial calcium promotes release of pro-apoptotic factors - again I can see no evidence for this in the paper. Calcium induced PTP opening in neurons tends to cause ATP depletion and necrotic cell death, and so I think the emphasis of cyt c release and apoptotic death should be reduced.

viii) Multiphoton FRET is tricky as excitation spectral profiles are often very broad. I struggled to see what optical arrangement was used for the calibration and it isn't clear whether the calibration was also done using multiphoton excitation - if so, this cannot be extrapolated safely to apply to the multiphoton measurements.

ix) It is stated in the text (line 83) that calibration is achieved simply by varying calcium conc. This alone is of course nonsensical. In searching the methods I discovered that the cells were permeabilised - this is critical and needs to be stated!

x) This group have previously shown calcium waves in astrocytes in this model. Is mitochondrial calcium also elevated in astrocytes?

xi) The relevance of the work for the human condition is slightly undermined if the components of the mitochondrial calcium signalling machinery are not changed in the mouse but are changed in man?

xii) In this context the paper by the groups of Bading and Hardingham (Nat Commun. 2013;4:2034) should be cited showing downregulation of MCU complex in response to glutamate exposure.

xiii) The implication of the disappearing mitochondria described (lines 234/5) is that the PTP is opening, causing loss of potential, loss of calcium and cell death. This is also consistent with the previous work from these authors showing the absence of mitochondria close to plaques. It would seem logical to test this in cyclosporin A treated mice? This seems to me very important to establish the PTP as a therapeutic target in AD and would greatly enhance the relevance of the paper.

xiv) It seems also necessary to refer to previous literature showing mitochondrial calcium uptake and/or PTP opening in relation to amyloid toxicity – see :: FASEB J. 2019 Jun;33(6):6697-6712; Biochem Soc Trans. 2018 Aug 20;46(4):829-842; PLoS One. 2017 Jan 6;12(1):e0168840; Biochem Biophys Res Commun. 2017 Feb 19;483(4):1110-1115; Nat Med. 2008 Oct;14(10):1097-105; Biosci Rep. 2001 Dec;21(6):789-800; J Neurosci. 2003 Jun 15;23(12):5088-95.

xv) Logically it is hard to reconcile a reduced mitochondrial membrane potential with increased calcium content unless the loss of potential is caused by the calcium overload – again this implicates PTP opening. My feeling is that the paper becomes much stronger if the authors can provide direct evidence for a role of the PTP in Abeta induced neuronal death, and this is (relatively speaking) not so hard for these authors to do.

Reviewer #3:

Remarks to the Author:

Bacskaï's group had used a ratiometric calcium indicator YC3.6 to demonstrate cytosolic calcium overload in the vicinity of individual plaques in APP/PS1 mice in vivo. In the current study, this group used mitochondria-targeted YC3.6 to detect mitochondrial calcium levels in the same APP/PS1 mouse model. After calibration of the calcium indicator in vitro and in vivo, they found mitochondrial calcium overload in a small fraction of the total mitochondria, independent of the distance from plaques, in 9 month old APP/PS1 mice but not in age-matched controls or younger APP/PS1 mice. They also found that those cells with higher mitochondrial calcium overload disappeared after 24 hrs. They further demonstrated that topical application of Abeta oligomers caused a similar mitochondrial calcium overload in WT mice which is dependent on MCU. Lastly, they compared the transcript levels of mitochondrial calcium related genes between AD and control using public database and found compensatory changes in the expression of these genes to avoid mitochondrial calcium overload in AD. They concluded that Abeta accumulation provokes MCU-dependent mitochondrial calcium overload in vivo which leads to neuronal death. This is a well executed study that provided convincing evidence of mitochondrial calcium overload in AD mouse models. There are several concerns:

- 1) The authors noted dramatic changes in the mitochondrial shape and size after topical application of Abeta. Does change in mitochondrial shape and size directly impact the measurement of YC3.6?
- 2) Presumably, mitochondrial calcium overload is secondary to cytosolic calcium overload. It is difficult to understand why cytosolic calcium overload occurs in the vicinity of plaques as the authors reported in their prior studies while mitochondrial calcium overload is independent of plaque distance. Shouldn't Abeta (or TgCM) also induce cytosolic calcium overload in Wt brain?
- 3) Both neuronal soma and neurite mitochondrial calcium can be measured in this study. Is there difference in the soma and neuritic mitochondrial calcium between Tg and control mice or in response to TgCM treatment?

4) The authors found disappearance of a small fraction of neurons with high mitochondrial calcium overload in their soma disappeared 24 hrs later. This appeared to be based on the loss of YFP/CFP signal rather than any direct evidence of loss of neurons. This is the major conclusion of the study. Given the importance of such conclusion, it should be strengthened. It is not clear whether there is neuronal loss in the APP/PS1 mice at 9 months of age. It appears that the topical application of Abeta (or TgCM) induced similar mitochondrial calcium overload in WT mice, was there similar loss of neurons in those neurons with more severe mitochondrial calcium overload?

5) The authors hypothesized that the oligomeric Abeta likely caused mitochondrial calcium overload. Soluble oligomeric abeta should be measured in the APP/PS1 mice at the two ages (2.5-3 months and 9 months) used in this study.

Reviewers' comments:

Reviewer #1 (Remarks to the Author):

This manuscript investigates the calcium handling of neuronal mitochondria in amyloid-driven mouse models of Alzheimer's disease (AD). There is a well-documented mitochondrial pathology associated with AD and it has already been shown in neuronal cultures that application of amyloid or expression of disease-associated variants of the amyloid pathway can lead to abnormal calcium handling in mitochondria. This study extends those findings to an in vivo context using a virally-delivered fluorescent calcium reporter to neuronal mitochondria in the cortex of mice – multiphoton microscopy was used to visualize the calcium sensor. The approach is an attractive one and, of course, it is important to know how and when mitochondrial biology goes awry in AD pathology. However, in this study, the major finding is that mitochondrial calcium handling is aberrant in an amyloid-dependent manner. This largely recapitulates what has been demonstrated in vitro, albeit in the living brain. There are some attempts to address how mitochondrial calcium goes wrong and what the consequences might be, but these mechanisms are not fully explored. For example, what links amyloid to mitochondrial calcium handling? what are the consequences of stopping calcium overload for pathology in the transgenic mice? Furthermore, there are a substantial number of specific concerns (detailed below) about methods and interpretation that need addressing. These uncertainties limit my enthusiasm for publication of the manuscript as it is.

Specific points:

Figure 1C is not at all convincing. There is so much COX-IV staining (naturally) that it would be difficult to rule out false colocalization, yet, in fact, the image makes it seem that there are locations with YC3.6 fluorescence (green) that are not COX-IV (red) positive. Perhaps some quantification, or a different imaging approach would help.

We thank the reviewer for the suggestion. Accordingly, Figure 1C has been replaced. We have performed a different immunohistochemistry using HSP60 (protein that localizes to the mitochondrial matrix) instead of COX-IV. In addition, Figure 1C shows intensity profile of ROIs across one of the cells, demonstrating that AAV.hSyn.2mtYC3.6 and HSP60 colocalize in mitochondria.

It is argued that the mitochondrial environment may change the properties of the YC3.6 - I'd agree that this could well be the case given the sensitivity of fluorescent proteins to redox state, pH and oxygen availability. However, these properties could equally be affected in AD neurons. So, how do we know that the change in FRET ratio is not an effect of differential mitochondrial environment in AD mice, potentially independent of calcium. A calibration of YC3.6 in AD neurons would confirm the way it performs.

Calibration of 2mtYC3.6 was performed in N2a cells. We selected YC3.6 because the YFP/CFP ratio does not change over a physiological range of pH (from 6.5 to 8.2), and gives a large Ca^{2+} dependent response that overwhelms the noise due to the pH change, therefore resulting in an excellent signal-to-noise ratio (Nagai T et al., *Proceedings of the National Academy of Sciences of the United States of America* 101, 10554-10559 (2004)). When expressed in mitochondria (vs cell body) the calibration curve shifted – in both WT and APP mice (and in cells in culture).

This could be related to the new genetic construct, or a different chemical environment in mitochondria. However, this observation highlights the advantage of ratiometric indicators as these effects can be ruled out with an appropriate re-calibration.

It is not clear in the methods if (and how) mice were anaesthetized for imaging – details are required. How is anesthesia depth monitored? is it the same in WT and APP mice? this is crucial as we know anaesthesia depth will affect neuronal activity and that activity will influence calcium in the cells and mitochondria.

We thank the reviewer for the comment. The induction method and monitoring during imaging (that was the same for every mouse) has been added to the Methods section as follows: “*Mice were anesthetized with isoflurane (5% inhalation, mixed with pure O₂) for induction, and a reduced concentration of isoflurane (~1.5%) was used during the imaging. The animal’s body temperature was maintained at ~37.5 °C during the imaging session with a heating pad; ophthalmic ointment was applied to protect the eyes. Mice were visually monitored throughout the duration of the imaging session (which lasted on average 2 hours)*”.

Figure 2E – the definition of Ca overload appears to be largely arbitrary. And the analysis of it is a simple recapitulation of the data in Figure 2C & 2D. It only really makes sense to define Ca overload if that level relates to some kind of mitochondrial pathology, irreversibility of the Ca overload or perhaps neuronal activity.

We agree with the reviewer that this is a somewhat arbitrary number, however, Ca²⁺ overload is defined using an unbiased statistical approach to directly compare absolute calcium levels between different cells/compartments (with ratiometric probes). Ca²⁺ overload was defined as YFP/CFP ratio greater than two standard deviations (SD) above the mean YFP/CFP ratio obtained for all the mitochondria in all Wt mice. This value has been used systematically in previous publications, including: Kuchibhotla KV, *et al.* Abeta plaques lead to aberrant regulation of calcium homeostasis in vivo resulting in structural and functional disruption of neuronal networks. *Neuron* 59, 214-225 (2008); Arbel-Ornath M, *et al.* Soluble oligomeric amyloid-beta induces calcium dyshomeostasis that precedes synapse loss in the living mouse brain. *Molecular neurodegeneration* 12, 27 (2017); Kastanenko KV, *et al.* Immunotherapy with Aducanumab Restores Calcium Homeostasis in Tg2576 Mice. *The Journal of neuroscience: the official journal of the Society for Neuroscience* 36, 12549-12558 (2016). These references have been added to the manuscript to support the definition of Ca²⁺ overload.

Figure 2D - The assertion that “only a small fraction of the total number of mitochondria exhibited elevated Ca²⁺” appears to be at odds with the very similarly-shaped distributions. Indeed, the curves in Figure 2D suggests that there is a general shift in Ca across the population. As suggested by the reviewer, this sentence has been removed from the text.

In many AD mouse models, neuronal activity has been shown to be perturbed. If neurons are firing differently then could the change in mitochondrial calcium taken as a snapshot not simply reflect that? Is the calcium overload in specific neurons or mitochondria (e.g. Figure 2B) persistent over time (i.e. can it be seen over multiple imaging sessions?)?

We appreciate this comment by the reviewer. While it is true that neurons fire and mitochondria are likely buffering the calcium increase in the cytosol, the basal levels (the steady levels at which the cell/mitochondria will go back after the increase) might be stable. To solve the

question of whether or not the levels of calcium are persistent over time, we have measured longitudinally (over 3 different imaging sessions/48h) the absolute mitochondrial calcium levels (ratiometric) in the same cells. New Figure 7a shows that during this period of time, static levels of mitochondrial calcium within the same cell were maintained. That was true for both low and higher calcium levels. However, extremely high levels of calcium in mitochondria (a very rare event) led to cell death, which is shown in Figure 7b.

For topical application of A β (Figure 4), it is not clear why young (4-5 months) mice were used compared to the effects on older (9 months) mice in transgenic model. Please justify and explain if this makes a difference when comparing experimental regimes.

The purpose of these set of experiments was to observe the effects of A β on the healthy leaving brain. 4-5 months old mice were used for technical reasons, because they are easily available from Jackson Labs at this age, and much less expensive. We also have not observed an age dependent effect in WT mice. See, for example, the data in Figure 2B. In addition, there is no reason to wait for 8-9 months of age (to deposit plaques) as for the APP/PS1 mice.

The block of A β -induced increase in mitochondrial calcium by Ru360 does suggest that the MCU is involved. However, it is possible that Ru360 renders neurons completely insensitive to A β , perhaps by altering their activity (and plasma membrane calcium entry pathways), or even damaging/killing the cells. It would be good to know that the neurons pre-treated with Ru360 are still functional in some sense.

It has been thoroughly described in many studies that Ru360 is a specific and potent inhibitor of MCU (Kirichok Y, Krapivinsky G, Clapham DE. *Nature* 427, 360-364 (2004); Matlib, M.A., et al. 1998. *J. Biol. Chem.* 273, 10223), and without affecting cell activity (Sanchez, J.A., et al. 2001. *J. Physiol.* 536, 387.). To confirm that the neurons were still responsive to A β after application of TgCM, we have performed the same experimental design (application of TgCM in presence or absence of Ru360), but using a calcium reporter targeted to the cytosol, using CBA.YC3.6 (Kuchibhotla KV, et al. *Neuron* 59, 214-225 (2008)). Supplemental Figure 5a shows representative pseudocolor images of the same fields of view imaged before and after application of either WtCM, TgCM, TgCM+Ru360 or Ru360 alone. Quantification in Figure 5b shows that calcium increased in the cytosol after application of TgCM in the presence of Ru360, suggesting that Ru360 did not render neurons insensitive to A β or blocked channels/influx pathways in the plasma membrane.

Figure 6 and Supplementary Figure 3 - The mitochondrial morphology analysis following A β application is underdone. It is not clear from the images shown how shape would actually be determined given the density of staining and whether the data in Supplementary Figure 3 relates specifically to mitochondria showing calcium overload (this would be a prediction perhaps?). Apparent changes in size could be manifest from small structures like mitochondria purely due to an increase in the fluorescence signal (were both CFP and YFP channels used in the size analysis?). Were similar morphological changes seen in the transgenic mice? Neuronal mitochondria have very different structure in axons and dendrites – can the morphology data be related to any particular type/location?

The reviewer is correct to notice that viral expression is very dense, so analyzing the complete z-stack was nearly impossible. To overcome this issue, mitochondrial size and shape were analyzed in the following manner: Approximately the same slice across the 3 times (basal, 1h

and 8h) was analyzed. A unique slice was taken per z-stack. ROIs were drawn with the free hand selection tool in ImageJ based in the pseudocolor images. All the images were acquired with the same settings, and brightness and contrast were maintained so that it did not affect the fluorescence signal (and therefore the size of the structures). About 100-125 random individual mitochondria were taken. Unfortunately, we could not differentiate between axons and dendrites. To overcome this problem, we tried to evaluate the mitochondria in the neurites at the same depth after application of both WtCM and TgCM (30-60 μm). New Supplemental Figure 6 shows how the specific ROIs were taken. Also, a better clarification of the procedure to analyze mitochondrial size and shape was added to the Methods section, as follows: *“To assess changes in mitochondrial morphology, a unique slice per z-stack was selected, matching the region across the 3 different time points (basal, 1h, 8h). Slices were selected within 30-60 μm depth. ROIs were drawn around individual mitochondria with the free hand selection tool in ImageJ based in the pseudocolor images. About 100-125 random individual mitochondria were taken. Matlab was used to measure the parameters of interest (area, perimeter and circularity) for each ROI”*.

Interestingly, data in Supplemental Figure 7 (previous Supplementary Figure 3) does not specifically relate to mitochondria showing calcium overload. As shown in the new scatter diagrams included in Supplemental Figure 7 (previous Supplemental Figure 3), there was no correlation between the mitochondrial calcium concentration and any of the parameters evaluated after WtCM application at any time point (basal, 1h or 8h), and 1h after TgCM application. There was negative although very weak linear correlation between $[\text{Ca}^{2+}]_{\text{mit}}$ and area, perimeter or circularity 8h after application of TgCM, suggesting a higher calcium concentration in mitochondria and smaller size.

Regarding the changes in transgenic mice, it was previously observed that there is mitochondrial loss and structural abnormalities in the same mouse model, including mitochondria swelling and fragmentation (Xie et al. J Neurosci 2013 **33**, 17042-17051). This has been added to the discussion section as follows: *“In addition, the same APP/PS1 Tg mouse model presents structural abnormalities in mitochondria, including mitochondria swelling and fragmentation. Along these lines, impaired balance of mitochondrial fission and fusion proteins has been shown in tissue from AD patients, and A β -induced mitochondrial fragmentation has been suggested to precede neuronal cell death”*.

Figure 7 – it is important to establish that this neuronal death effect is specific to the AD transgenic mice (as opposed to a non-specific viral or protein expression effect). As such, the WT control is an important omission.

Interestingly, the effect of high calcium in mitochondria leading to cell death is a process that would take place in any neuron with high mitochondrial calcium levels (even in Wt mice). However, it is a very rare event even in APP mice, and almost never observed in Wt mice. It is therefore much more likely to happen in Tg mice, since mitochondrial calcium levels are increased in this group.

The relevance of the human data analysis to the rest of the study is loose at best. We already know that AD is associated with mitochondrial defects and these analyses confirm there are

changes. However, without further experiments to link the human data to the mouse findings, Figure 8 should not be part of this manuscript.

The reviewer is correct that mitochondrial dysfunction has been largely described in AD. However, in this study we show for the first time that there are alterations in gene expression in the human brain of proteins related to mitochondrial calcium homeostasis (both influx and efflux), which provides strong translational relevance to our work in mice. We are aware that the results are unexpected, but as we state in the discussion section, we believe that this is a compensatory effect of mitochondria to prevent the calcium overload observed *in vivo*. In addition, human data (RNA-seq from hundreds of patients) makes the study more translational and powerful. For these reasons, we consider that data from human RNA-seq (Figure 8) should be kept in the manuscript. However, previous Figure 8 is now New Supplemental Figure 9.

Line 72 - “YC3.6 is sufficiently bright and exhibits a large dynamic range, resulting in an excellent signal-to-noise ratio.” - This needs to be more specific (“sufficiently bright” for what?) and needs to refer to published data or data contained in this manuscript.

We thank the reviewer for the suggestion. The sentence has been rephrased to: “YC3.6 is one of the brightest reporters among the YC versions, and exhibits a large dynamic range, resulting in an excellent signal-to-noise ratio”; and the following reference has been added: Nagai T, Yamada S, Tominaga T, Ichikawa M, Miyawaki A. Expanded dynamic range of fluorescent indicators for Ca(2+) by circularly permuted yellow fluorescent proteins. *Proceedings of the National Academy of Sciences of the United States of America* 101, 10554-10559 (2004).

Reviewer #2 (Remarks to the Author):

Thank you for asking to me to read this interesting paper. The authors have used a genetically encoded calcium sensitive reporter targeted to neuronal mitochondria to explore changes in mitochondrial calcium content and signalling in a transgenic mouse overexpressing Abeta, show that there is an increased mitochondrial calcium concentration in neurons close to amyloid plaques, and argue that this might drive progression to cell death. In the latter part of the work the authors have explored the expression levels of protein components of the mitochondrial calcium influx and efflux pathways and – rather oddly, perhaps - there is no change in the mouse but that show that, in genetic databases from human material, the components of the MCU complex are downregulated while the NLCX is upregulated. The concept is not novel but the experimental data in an *in vivo* model is potentially very interesting.

The work is nicely presented, and seems to have been well done as one would expect from this excellent lab., and follows on from a number of papers from this lab along a rather similar theme. I do need to raise some issues however.

i) In order to interpret the meaning of a rise in mitochondrial calcium content it is critical to know what is happening to cytosolic calcium signals. The mitochondrial signal follows the cytosolic signal - if the cytosolic calcium signal is elevated it is not surprising if the mitochondrial signal is also elevated. If the cytosolic signal is normal, then a rise in the

mitochondrial signal appears as a more specific phenomenon driving the pathophysiology. It is very hard to judge whether this relationship has changed in any way as the only comparator is an old paper from the same group that showed (I believe) a very modest increase in cytosolic calcium content in neurons close to the plaques. To understand this better, it would be best to look at dynamic changes in cytosolic and mitochondrial calcium in the same cells if that is possible and ask whether this changes in the disease models.

Unfortunately, we have not been able to observe calcium changes in cytosol and mitochondria simultaneously in vivo with multiphoton microscopy and ratiometric reporters yet, since the tools available to the research community do not allow to multiplex with two photon microscopy (due to low brightness, low dynamic range, fast photo bleaching and spectral crosstalk).

Therefore, we have addressed this question in vitro: we have used cortical primary neurons loaded with the dye calbryte 590 (reports Ca^{2+} changes in the cytosol, emission in the red channel) and infected with our genetically encoded calcium reporter (hsyn.2mtYC3.6) (reports Ca^{2+} changes in mitochondria, ratiometric emission with two photon) and we have stimulated them with either KCl or TgCM. We have been able to observe calcium rises in the cytosol and in mitochondria simultaneously. Data shows that both KCl and TgCM were able to simultaneously increase $[\text{Ca}^{2+}]_{\text{cyt}}$ and $[\text{Ca}^{2+}]_{\text{mit}}$. However, both graphs show that mitochondria exhibit different kinetics: there is a synchronous transient increase of $[\text{Ca}^{2+}]_{\text{mit}}$ and $[\text{Ca}^{2+}]_{\text{cyt}}$ after the stimulus is applied, but the mitochondrial Ca^{2+} remains elevated, maybe because the extrusion system acts slower than cytosolic Ca^{2+} clearance (after application of KCl). Also, after TgCM application $[\text{Ca}^{2+}]_{\text{mit}}$ kept rising, whereas $[\text{Ca}^{2+}]_{\text{cyt}}$ increase is slower. We are also aware that these differences in kinetics could be due to differences in the Ca^{2+} affinities of the two Ca^{2+} sensors used. These data are shown in Supplemental Figure 2. In addition to this, it has been previously demonstrated that inhibiting calcium overload in mitochondria (by a mild depolarization) induced by amyloid beta oligomers prevents from cell death despite the increase in cytosolic calcium (Sanz-Blasco, PlosONE 2008 and Calvo-Rodriguez, J Alzheimers Dis 2016), suggesting that mitochondrial calcium plays a key role in the neurotoxicity induced by amyloid beta. The following paragraph has been added to the Results section: *“We also compared the kinetics of cytosolic Ca^{2+} versus mitochondrial Ca^{2+} in primary cells stimulated with KCl. We used cortical neurons loaded with the dye calbryte 590 and infected with our mitochondrial reporter hSyn.2mtYC3.6, to measure Ca^{2+} changes in the cytosol and mitochondria simultaneously (Supplementary Figure 2). Application of KCl increased both cytosolic Ca^{2+} concentration ($[\text{Ca}^{2+}]_{\text{cyt}}$) and $[\text{Ca}^{2+}]_{\text{mit}}$. However, mitochondria exhibit different kinetics, which justified the need to address the mitochondrial calcium impairment independent of cytosolic calcium in the disease model”.*

ii) Showing a rise in mitochondrial calcium in response to 300mM (?) KCl used as shown doesn't really tell us very much except that the probe works. At the very least I am surprised that the authors haven't asked how this response changes in the disease model.

As the reviewer was anticipating, the only purpose of the application of KCl in vivo was just to show that the calcium reporter is functional and provides calcium changes in vivo.

iii) 300mM KCl will surely be very hypertonic?

Usually the concentrations of probes applied directly to the brain are higher than what is applied in vitro. That is the reason of applying KCl 300 mM. This concentration is rapidly diluted in the cortex and cleared from parenchyma, so it is difficult to determine the exact concentration

neurons are exposed to, and for how long. In addition, a video has been uploaded to probe that the cells are acting normal (and not shrinking with time). Again, within 30 minutes after KCl application, the calcium levels were back to normal and KCl could be reapplied.

iv) Previous work from this group suggested that mitochondrial membrane potential is selectively reduced in neurons close to plaques. They have also shown an absence of mitochondria close to plaques which makes the present work even more complex to interpret. As mitochondrial calcium uptake is potential dependent, the observations become rather surprising and need explanation. What is driving a rise in mitochondrial calcium signal despite a reduced potential? Or is the reduced potential driven by the increased mitochondrial calcium content and PTP opening? This is experimentally testable either by chelating calcium or by inhibiting the PTP.

We appreciate the comment by the reviewer. In order to understand the effects of soluble A β on mitochondrial calcium and mitochondrial membrane potential ($\Delta\Psi_m$), we have used primary neurons and, as suggested by the reviewer, exposed them to TgCM (soluble oligomeric A β) in the presence (or absence) of cyclosporine A (CsA), which is a drug able to block the mPTP. First we observed that TgCM was able to increase mitochondrial calcium in primary cells (infected with hSyn.2mtYC3.6), and to depolarize mitochondria (measured with TMRE). Pre-incubation and co-application of TgCM with CsA was able to block the mPTP (measured with calcein/cobalt fluorescence) and to prevent mitochondrial depolarization, but not to inhibit the mitochondrial calcium overload elicited by TgCM. These data suggest that the reduction in mitochondrial potential is driven by the increase in mitochondrial calcium after TgCM application. These results are shown in New Supplemental Figure 8, and the following paragraph has been added to the Results section: *“In order to further understand the effects of increased mitochondrial Ca²⁺ concentration and the connection to neuronal cell death, we performed some mechanistic studies in vitro using primary cortical neurons exposed to TgCM. First, we determined that TgCM increased [Ca²⁺]_{mit} in primary neurons (infected with hSyn.2mtYC3.6) (Supplemental Figure 8A) within 1h, corroborating the in vivo data. In addition, TgCM was able to decrease the mitochondrial membrane potential ($\Delta\Psi_m$) (measured with TMRE) in primary neurons (Supplemental Figure 8D) and to activate the mPTP as evaluated by calcein/cobalt fluorescence (Supplemental Figure 8C). Finally, we evaluated the effects of TgCM on neuronal apoptosis by caspase activation. TgCM activated caspases 3/7 (detected with CellEvent Caspase 3/7 green detection reagent) in the primary neurons (Supplemental Figure 8E). Interestingly, these effects took place in the same window of time as the increase of [Ca²⁺]_{mit}. WtCM did not drive any of these effects on mitochondria”;* and in the Discussion section: *“Furthermore, our in vitro data show that soluble A β increases mitochondrial Ca²⁺, leading activation of mPTP, $\Delta\Psi_m$ decrease and activation of caspases. Altogether, these observations suggest that mitochondrial Ca²⁺ plays a key role in the neurotoxicity induced by A β ”.*

v) I understand that the calcium reporter should be specific for neurons (this should be stated more clearly) but perhaps some immunofluorescence should confirm that it is selectively expressed in neurons?

We thank the reviewer for this suggestion. The word “neurons” and “neuronal” has been added in the text when referring to the calcium reporter. In addition, we have performed an ex vivo immunofluorescence to prove colocalization of hsyn.2mtYC3.6 with NeuN (marker for neurons) and not with GS (marker for astrocytes): A C57 mouse was intracranially injected with

hsyn.2mtYC3.6, and after 3 weeks to allow for expression the mouse was euthanized and the brain extracted and sliced with a cryostat. Immunofluorescence against GFP, NeuN and GS was performed. This can be observed in New Supplemental Figure 1.

vi) The final scheme appears to suggest that Abeta promotes increased calcium through voltage gated calcium channels – I can't see here any evidence to support that model.

The graphical abstract has been changed according to reviewer suggestion.

vii) The final scheme also suggests that the rise in mitochondrial calcium promotes release of pro-apoptotic factors – again I can see no evidence for this in the paper. Calcium induced PTP opening in neurons tends to cause ATP depletion and necrotic cell death, and so I think the emphasis of cyt c release and apoptotic death should be reduced.

The graphical abstract has been changed according to reviewer suggestion. Also, text in the discussion has been changed so that the references to cyt c release and apoptotic cell death are reduced.

viii) Multiphoton FRET is tricky as excitation spectral profiles are often very broad. I struggled to see what optical arrangement was used for the calibration and it isn't clear whether the calibration was also done using multiphoton excitation – if so, this cannot be extrapolated safely to apply to the multiphoton measurements.

The calibration was performed in N2a cells transduced with the same viral construct used in vivo using the same Olympus FluoView FV1000MPE multiphoton laser-scanning system mounted on an Olympus Bx61WI microscope, and with the same settings as for the in vivo acquisition.

These specifications have been added to the Methods section as follows: “*2 photon excitation was used at 860 nm, and emitted light was collected in three channels in the range of 380-480, 500-540 and 560-650 nm. Images were acquired on an Olympus FluoView FV1000MPE multiphoton laser-scanning system mounted on an Olympus Bx61WI microscope, and photomultiplier settings were selected and maintained for the intravital imaging*”.

ix) It is stated in the text (line 83) that calibration is achieved simply by varying calcium conc. This alone is of course nonsensical. In searching the methods I discovered that the cells were permeabilised – this is critical and needs to be stated!

We appreciate the comment made by the reviewer. A sentence stating that cells were permeabilized has been added to the main text: “*We then calibrated hsyn.2mtYC3.6 in N2a cells. Cells were first transfected with hSyn.2mtYC3.6. 24 h later, cells were permeabilized and then exposed to solutions containing increasing known $[Ca^{2+}]$ (See Methods)*”.

x) This group have previously shown calcium waves in astrocytes in this model. Is mitochondrial calcium also elevated in astrocytes?

This is a really interesting question that we are currently trying to address. The calcium waves in astrocytes are incredibly rare, making them difficult to probe. And it is likely that we would need to detect them with intracellular calcium imaging simultaneously with mitochondrial imaging and the spectral overlap and/or limitations of existing probes makes this difficult if not impossible in vivo. However, our goals are to try and attempt this, although we consider that this would belong to a different study.

xi) The relevance of the work for the human condition is slightly undermined if the components of the mitochondrial calcium signalling machinery are not changed in the mouse but are changed in man?

We agree with the reviewer that the observation of no changes in mouse is slightly disappointing. However, we considered it relevant to show the changes of the genes encoding proteins related to mitochondrial calcium influx and efflux in the human AD brain. As we state in the discussion section, we believe that this is a compensatory effect of the human mitochondria to prevent the calcium overload observed *in vivo*. In addition, human data (RNA-seq from hundreds of patients) makes the study more translational and powerful. We consider that data from human RNA-seq (Figure 8) should be kept in the manuscript. However, previous Figure 8 is now New Supplemental Figure 9.

xii) In this context the paper by the groups of Bading and Hardingham (Nat Commun. 2013;4:2034) should be cited showing downregulation of MCU complex in response to glutamate exposure.

We thank the reviewer for the suggestion. Qiu et al., Nat Comm 2013 is now cited and their work discussed as follows: “*MCU has previously been related to excitotoxic neuron cell death, since overexpression of MCU in mitochondria in vitro increased $[Ca^{2+}]_{mit}$ following activation of NMDA receptors, leading to mitochondrial membrane depolarization and cell death. These previous results established MCU as a mediator of death signals-induced loss of mitochondrial membrane potential and as a therapeutic target for excitotoxicity*”.

xiii) The implication of the disappearing mitochondria described (lines 234/5) is that the PTP is opening, causing loss of potential, loss of calcium and cell death. This is also consistent with the previous work from these authors showing the absence of mitochondria close to plaques. It would seem logical to test this in cyclosporin A treated mice? This seems to me very important to establish the PTP as a therapeutic target in AD and would greatly enhance the relevance of the paper.

We refer here to responses IV and XV above and below. The results from the CsA treatment in *in vitro* primary neurons have been added and discussed in the text as follows: *On the other hand, our in vitro work shows that preventing activation of the mPTP secondary to mitochondrial Ca^{2+} overload could also be proposed as a therapeutic target for AD, since preventing mPTP activation protects mitochondria from depolarization (loss of $\Delta\Psi_m$) and subsequent activation of apoptosis. Previous studies have already shown a link between $A\beta$, mPTP and cell death in AD in neurons and astrocytes, and here we show that this event is secondary to mitochondrial Ca^{2+} overload, since preventing mPTP activation did not forestall mitochondrial Ca^{2+} overload driven by soluble $A\beta$, but avoided mitochondrial depolarization and caspase activation.*

xiv) It seems also necessary to refer to previous literature showing mitochondrial calcium uptake and/or PTP opening in relation to amyloid toxicity – see :: FASEB J. 2019 Jun;33(6):6697-6712; Biochem Soc Trans. 2018 Aug 20;46(4):829-842; PLoS One. 2017 Jan 6;12(1):e0168840; Biochem Biophys Res Commun. 2017 Feb 19;483(4):1110-1115; Nat Med. 2008 Oct;14(10):1097-105; Biosci Rep. 2001 Dec;21(6):789-800; J Neurosci. 2003 Jun 15;23(12):5088-95.

We thank the reviewer for the suggestion. This literature has been included in the text.

xv) Logically it is hard to reconcile a reduced mitochondrial membrane potential with increased calcium content unless the loss of potential is caused by the calcium overload – again this implicates PTP opening. My feeling is that the paper becomes much stronger if the authors can provide direct evidence for a role of the PTP in Abeta induced neuronal death, and this is (relatively speaking) not so hard for these authors to do.

We agree with the reviewer that the mPTP could be more strongly established as a therapeutic target in AD, by directly relating the mPTP to neuronal cell death induced by A β in vivo.

Unfortunately, this mouse model does not present a high number of neurons that die (this is an extremely rare event in this mouse model, as shown in the present study and in Xie et al, PNAS 2013). Therefore, it would be extremely challenging if not impossible to convincingly establish the direct connection of reduced neuronal cell death by blocking mPTP in this model. For this reason, we evaluated the effects of blocking mPTP (with CsA) and activation of the apoptosis pathway in primary neurons. We evaluated the effects of TgCM on neuronal apoptosis by caspase activation (with CellEvent Caspase 3/7 green detection reagent). Application of TgCM elicited activation of caspases in the primary neurons. Blocking the mPTP with CsA led to a trend in decreased numbers of neurons that led to caspases 3/7 activation. These data are shown in Supplemental Figure 8, and the following paragraph has been added to the Results section:

“Interestingly, blocking of the mPTP with cyclosporine A (CsA), as verified by calcein/cobalt fluorescence, prevented the decrease of $\Delta\Psi_m$, but did not inhibit the increase of $[Ca^{2+}]_{mit}$ elicited by TgCM. These results suggest that the reduction of mitochondrial membrane potential is driven by the increase in mitochondrial Ca^{2+} after TgCM application and not the opposite. Importantly, blocking the mPTP with CsA decreased the number of neurons that led to caspases 3/7 activation, albeit the reduction was not statistically significant (Supplemental Figure 8)”.

Reviewer #3 (Remarks to the Author):

Bacsikai's group had used a ratiometric calcium indicator YC3.6 to demonstrate cytosolic calcium overload in the vicinity of individual plaques in APP/PS1 mice in vivo. In the current study, this group used mitochondria-targeted YC3.6 to detect mitochondrial calcium levels in the same APP/PS1 mouse model. After calibration of the calcium indicator in vitro and in vivo, they found mitochondrial calcium overload in a small fraction of the total mitochondria, independent of the distance from plaques, in 9 month old APP/PS1 mice but not in age-matched controls or younger APP/PS1 mice. They also found that those cells with higher mitochondrial calcium overload disappeared after 24 hrs. They further demonstrated that topical application of Abeta oligomers caused a similar mitochondrial calcium overload in WT mice which is dependent on MCU. Lastly, they compared the transcript levels of mitochondrial calcium related genes between AD and control using public database and found compensatory changes in the expression of these genes to avoid mitochondrial calcium overload in AD. They concluded that Abeta accumulation provokes MCU-dependent mitochondrial calcium overload in vivo which leads to neuronal death. This is a well executed study that provided convincing evidence of mitochondrial calcium overload in AD mouse models. There are several concerns:

1) The authors noted dramatic changes in the mitochondrial shape and size after topical application of Aβ. Does change in mitochondrial shape and size directly impact the measurement of YC3.6?

The value of ratiometric reporters negates this concern, as it inherently corrects for changes or differences in concentration of the probe. However, we performed a correlation test between the $[Ca^{2+}]_{mit}$ (YFP/CFP ratio) with the area, perimeter and circularity. There was no correlation between the mitochondrial calcium concentration and any of the parameters evaluated after WtCM application at any time point (basal, 1h or 8h), and 1h after TgCM application. There was negative although very weak linear correlation between $[Ca^{2+}]_{mit}$ and area, perimeter or circularity 8h after application of TgCM, suggesting a higher calcium concentration in mitochondria and smaller size. The scatter diagrams are shown in the insets of Supplemental Figure 7.

2) Presumably, mitochondrial calcium overload is secondary to cytosolic calcium overload. It is difficult to understand why cytosolic calcium overload occurs in the vicinity of plaques as the authors reported in their prior studies while mitochondrial calcium overload is independent of plaque distance.

We thank the reviewer for the comment. We were also surprised that no dependence was found between mitochondrial Ca^{2+} overload and distance to amyloid plaques. These data might suggest that mitochondria could be sensing and buffering the high calcium concentration everywhere (close and far from plaques, due to oligomeric Aβ), whereas in order to observe an increase in Ca^{2+} concentration in the cytosol the neurons needed to be closer to the plaques, where the concentration of the oligomeric Aβ is believe to be higher. In addition, mitochondria are more mobile and can undergo mitophagy, which could contribute to this effect. Additionally, calcium overload in neurites is a rare event (even near plaques), and calcium overload in mitochondria is even more rare. Coupled with the likelihood that calcium overload in mitochondria might be a final step that leads to apoptosis, it is possible that severely affected neurites are rapidly pruned, diluting a local effect of plaques. We are aware that this is only speculation, and likely more mechanistic studies (such as multiplexing with ratiometric tools in the same cell) would be needed to resolve this question. This issue was already discussed in the text as follows: *“This is in contrast to cytosolic Ca^{2+} overload, wherein there was a higher likelihood of detecting Ca^{2+} overload in the immediate vicinity of individual plaques. This suggests that diffusible soluble Aβ oligomeric species rather than plaques are likely responsible for the increase of mitochondrial $[Ca^{2+}]$ in the neurons of the APP/PS1 mouse, or that mitochondria are less vulnerable than cytosol to the pathophysiological insult mediated by Aβ. The fact that mitochondria are dynamic entities that can move (or undergo fusion/fission) along the axon or that can be subjected to mitophagy when damaged could also contribute to this difference”*.

Shouldn't Aβ (or TgCM) also induce cytosolic calcium overload in Wt brain?

We recently demonstrated that soluble oligomeric Aβ (TgCm) increased calcium overload in the cytosol of neurons linked to synapse loss in the Wt brain (Arbel-Ornath M, et al. Soluble oligomeric amyloid-beta induces calcium dyshomeostasis that precedes synapse loss in the living mouse brain. *Molecular neurodegeneration* 12, 27 (2017)). In addition, we have repeated these experiments in this work to increase rigor and reproducibility by topically applying TgCM to the

brain of Wt mice expressing CBA.YC3.6 (targeting the cytosol of neurons) and have also observed an increase in the cytosolic Ca^{2+} . This is shown in Supplemental Figure 5.

3) Both neuronal soma and neurite mitochondrial calcium can be measured in this study. Is there difference in the soma and neuritic mitochondrial calcium between Tg and control mice or in response to TgCM treatment?

We have measured the mitochondrial calcium in these two different locations (somas and neurites) in the APP/PS1 Tg mice (and Wt control mice) at 9 months of age (now in Figure 2F), in the APP/PS1 Tg mice (and Wt control mice) at 3 months of age (now in Supplemental Figure 3E) and after application of WtCM and TgCM (now in Figure 4G). The differences observed between Wt and Tg APP/PS1 mice were similar to the differences observed when the total mitochondrial content was evaluated: $[Ca^{2+}]_{mit}$ in both somas and neurites was elevated in the Tg APP/PS1 at 9 months of age compared to Wt, and no differences were found at 3 months of age. Interestingly, TgCM increased $[Ca^{2+}]_{mit}$ only in neurites, possibly suggesting a more acute vulnerability to this insult.

4) The authors found disappearance of a small fraction of neurons with high mitochondrial calcium overload in their soma disappeared 24 hrs later. This appeared to be based on the loss of YFP/CFP signal rather than any direct evidence of loss of neurons. This is the major conclusion of the study. Given the importance of such conclusion, it should be strengthened. It is not clear whether there is neuronal loss in the APP/PS1 mice at 9 months of age.

In order to strengthen this conclusion, we have performed the same time lapse (imaging of the same field of view for 2 consecutive days) in healthy neurons (low mitochondrial Ca^{2+} levels) and in mitochondrial overloaded calcium neurons, concomitantly injected with Hoechst 33342 to observe the nucleus. This was a particularly challenging experiment but we agree that it would really strengthen the manuscript. We could observe that the nucleus from healthy cells was spherical and DNA was evenly distributed. On the contrary, in cells with high mitochondrial Ca^{2+} (likely undergoing apoptosis as also shown in vitro in new Supplemental Figure 8) the nucleus became condensed, in addition to the loss of the whole soma as marked by the loss of YFP/CFP fluorescence. This can be observed in New Figure 7C. It is important to point out that this is an extremely rare event in this mouse model, but this confirms previous results from this group where a tiny fraction of cell death events was also observed (Xie et al., PNAS 110, 7904-7909 2013), despite the well-established lack of overt neurodegeneration or atrophy in this mouse model.

It appears that the topical application of Abeta (or TgCM) induced similar mitochondrial calcium overload in WT mice, was there similar loss of neurons in those neurons with more severe mitochondrial calcium overload?

Even though application of TgCM to the Wt brain increased mitochondrial Ca^{2+} levels, it went back to normal levels after 8h. Topical application of anything soluble to the intact brain immediately diffuses and gets cleared through normal CSF turnover/clearance pathways. Therefore, it is almost impossible to monitor the concentration and time course of the treatment using this approach limiting our ability detect neuronal cell death. However, the treatment was sufficient to allow detectable changes in calcium. This question has already been discussed in the manuscript as follows: “*Application of TgCM to the cortical surface of the mouse brain increased $[Ca^{2+}]$ in neuronal mitochondria within 1h, whereas conditioned media from Wt*

littermates or Aβ-immunodepleted conditioned media did not. [...] Importantly, the TgCM-induced mitochondrial Ca²⁺ overload was reversible, as longitudinal in vivo imaging demonstrated its normalization 8h after TgCM application, implying extrusion via mitochondrial Ca²⁺ exchangers, and/or clearance of the Aβ from the brain parenchyma”.

5) The authors hypothesized that the oligomeric Abeta likely caused mitochondrial calcium overload. Soluble oligomeric abeta should be measured in the APP/PS1 mice at the two ages (2.5-3 months and 9 months) used in this study.

The mouse model used (APP/PS1) has been extensively characterized, and the levels of the soluble oligomeric abeta measured previously and compared at different ages by many groups including (Garcia-Alloza M et al., Characterization of amyloid deposition in the APP^{swe}/PS1^{dE9} mouse model of Alzheimer disease. *Neurobiology of disease* 24, 516-524 (2006); Takeda S, et al., Brain interstitial oligomeric amyloid β increases with age and is resistant to clearance from brain in a mouse model of Alzheimer's disease. *FASEB J.* 2013;27(8):3239–3248. doi:10.1096/fj.13-229666; Maia LF, et al., Changes in Aβ and tau in the cerebrospinal fluid of transgenic mice overexpression amyloid precursor protein. *Science Translational Medicine* 2013, 5:194re2; Manocha GD, et al., Temporal progression of Alzheimer’s disease in brains and intestines of transgenic mice. *Neurobiology of Aging* 2019, 81:166-176). We have added this information to the manuscript as follows: “*In addition, it has been shown that the concentration of soluble oligomeric Aβ in the brain of this mouse model increases with age, which could contribute to the increase in [Ca²⁺]_{mit} as the animal ages”.*

** See Nature Research's author and referees' website at www.nature.com/authors for information about policies, services and author benefits

This email has been sent through the Springer Nature Tracking System NY-610A-NPG&MTS

Confidentiality Statement:

This e-mail is confidential and subject to copyright. Any unauthorised use or disclosure of its contents is prohibited. If you have received this email in error please notify our Manuscript Tracking System Helpdesk team at <http://platformsupport.nature.com>.

Details of the confidentiality and pre-publicity policy may be found here <http://www.nature.com/authors/policies/confidentiality.html>

Privacy Policy | Update Profile

DISCLAIMER: This e-mail is confidential and should not be used by anyone who is not the original intended recipient. If you have received this e-mail in error please inform the sender and delete it from your mailbox or any other storage mechanism. Springer Nature America, Inc. does not accept liability for any statements made which are clearly the sender's own and not expressly made on behalf of Springer Nature America, Inc. or one of their agents. Please note that neither Springer Nature America, Inc. or any of its agents accept any responsibility for viruses that may be contained in this e-mail or its attachments and it is your responsibility to scan the e-mail and attachments (if any).

Reviewers' Comments:

Reviewer #1:

Remarks to the Author:

The manuscript has improved as a result of additions related to all the reviewers' comments, and my comments have been addressed by addition of explanatory text, remodelled figures and new experiments. I am pleased to see the longitudinal imaging in new Figure 7, which addresses queries about the longevity of the mitochondrial calcium rise. In addition, the new in vitro data implicating mPTP and caspase activation downstream of the calcium overload fills in some of the missing mechanistic links that were only inferred in the original manuscript. Of course, it would have been better to see these effects in the in vivo models that are the basis of the paper and to measure the (potentially) beneficial effects on the mice of intervening in them. So, these pathways remain tantalizing possibilities rather than definitive therapeutic targets. However, this is significantly more work, and I am inclined to think that the manuscript will already be of interest to many, and so should be published.

Reviewer #2:

Remarks to the Author:

The revised manuscript is improved over the original submission. I'm afraid that I am a little disappointed that the mechanism of increased mitochondrial calcium content in this model is still not very clear :

One would normally expect that if cytosolic calcium is elevated, mitochondrial calcium will also be elevated and so there is nothing very surprising there, and so as I argued before, what might be really interesting is to ask whether the relationship between cytosolic calcium content and mitochondrial calcium is altered. I appreciate the technical issues in combining these measurements although I'd have thought that the use of conventional chemical calcium indicators might be possible. That the kinetics of mitochondrial and cytosolic calcium signals are different is not surprising at all and has been described in many different model systems. It does seem odd, however, in this regard that elevated cytosolic calcium is described closer to plaques, while the mitochondrial signal seems less closely related to distance from plaques. I don't really understand how this would work, and the discussion in the rebuttal seems to miss the point.

I also have some relatively minor issues:

I have a quarrel with the statement that .. 'blocking the mPTP with CsA decreased the number of neurons that led to caspases 3/7 activation, albeit the reduction was not statistically significant ...). ' It seems to me that this is really not acceptable - if the result is not significant, the difference could have arisen by chance and therefore it is not 'Importantly' and cannot really be presented as useful data in a paper like this.

You also write on page 11: 'These results suggest that the reduction of mitochondrial membrane potential is driven by the increase in mitochondrial Ca²⁺ after TgCM application and not the opposite.' I don't understand what you mean - the 'opposite' would mean that a rise in mitochondrial calcium is driven by a fall in mitochondrial membrane potential, which simply doesn't happen.

On p8, line 178 it would be interesting to know the time course of the increase in mitochondrial calcium after application of the TgCM (Fig 4).

Reviewer #3:

Remarks to the Author:

The measurement of Hoechst 33342 signal helped strengthen the claim that the disappearing cells are likely dying, but the representative picture shown in Figure 7C is not very convincing to demonstrate that this dying/disappearing neuron had calcium overload. It will be great if a better picture can be shown.

REVIEWERS' COMMENTS:

Reviewer #1 (Remarks to the Author):

The manuscript has improved as a result of additions related to all the reviewers' comments, and my comments have been addressed by addition of explanatory text, remodeled figures and new experiments. I am pleased to see the longitudinal imaging in new Figure 7, which addresses queries about the longevity of the mitochondrial calcium rise. In addition, the new in vitro data implicating mPTP and caspase activation downstream of the calcium overload fills in some of the missing mechanistic links that were only inferred in the original manuscript. Of course, it would have been better to see these effects in the in vivo models that are the basis of the paper and to measure the (potentially) beneficial effects on the mice of intervening in them. So, these pathways remain tantalizing possibilities rather than definitive therapeutic targets. However, this is significantly more work, and I am inclined to think that the manuscript will already be of interest to many, and so should be published.

We thank the reviewer for the positive comments.

Reviewer #2 (Remarks to the Author):

The revised manuscript is improved over the original submission. I'm afraid that I am a little disappointed that the mechanism of increased mitochondrial calcium content in this model is still not very clear :

One would normally expect that if cytosolic calcium is elevated, mitochondrial calcium will also be elevated and so there is nothing very surprising there, and so as I argued before, what might be really interesting is to ask whether the relationship between cytosolic calcium content and mitochondrial calcium is altered. I appreciate the technical issues in combining these measurements although I'd have thought that the use of conventional chemical calcium indicators might be possible. That the kinetics of mitochondrial and cytosolic calcium signals are different is not surprising at all and has been described in many different model systems. It does seem odd, however, in this regard that elevated cytosolic calcium is described closer to plaques, while the mitochondrial signal seems less closely related to distance from plaques. I don't really understand how this would work, and the discussion in the rebuttal seems to miss the point.

We agree with the reviewer that we can just speculate the reasons underlying this difference with the tools that we have in hand as of today to address this complicated question in vivo. For further explanation, we have rephrased our sentence, which now states: “*The intrinsic properties of mitochondria with regards to \$Ca^{2+}\$ kinetics, and as dynamic entities that undergo axonal transport, fusion/fission, and mitophagy when damaged, could also explain this difference*”.

I also have some relatively minor issues:

I have a quarrel with the statement that .. 'blocking the mPTP with CsA decreased the

number of neurons that led to caspases 3/7 activation, albeit the reduction was not statistically significant ...). ' It seems to me that this is really not acceptable – if the result is not significant, the difference could have arisen by chance and therefore it is not 'Importantly' and cannot really be presented as useful data in a paper like this.

We acknowledge the reviewer's comment, and we have rephrased the sentence to: *“Blocking the mPTP with CsA showed a trend towards decreasing the number of neurons that led to caspases 3/7 activation (Supplementary Figure 8)”*.

You also write on page 11: 'These results suggest that the reduction of mitochondrial membrane potential is driven by the increase in mitochondrial Ca²⁺ after TgCM application and not the opposite.' I don't understand what you mean – the 'opposite' would mean that a rise in mitochondrial calcium is driven by a fall in mitochondrial membrane potential, which simply doesn't happen.

We agree that this statement is unclear, and we appreciate that the reviewer pointed this out. The sentence now states: *“These results suggest that the reduction of mitochondrial membrane potential is driven by the increase in mitochondrial Ca²⁺ after TgCM application”*.

On p8, line 178 it would be interesting to know the time course of the increase in mitochondrial calcium after application of the TgCM (Fig 4).

We agree that it could be interesting to follow the more acute time course, but our previous experiences using conditioned media suggest that responses in the brain are relatively slow, are very different than what happens in cell culture experiments, and are adequately captured at the 1 hr time point.

Reviewer #3 (Remarks to the Author):

The measurement of Hoechst 33342 signal helped strengthen the claim that the disappearing cells are likely dying, but the representative picture shown in Figure 7C is not very convincing to demonstrate that this dying/disappearing neuron had calcium overload. It will be great if a better picture can be shown.

We thank the reviewer for the comment. In addition, Figure 7C has been modified so the cells can be better appreciated.

** See Nature Research's author and referees' website at www.nature.com/authors for information about policies, services and author benefits

This email has been sent through the Springer Nature Tracking System NY-610A-NPG&MTS